# Uncovering the Redundancy in Transformers via a Unified Study of Layer Dropping

**Shwai He**[*], **Guoheng Sun**[*], **Zheyu Shen, Ang Li**
*University of Maryland, College Park*
*{shwaihe, ghsun, zyshen, angliece}@umd.edu*

**Reviewed on OpenReview:** *https://openreview.net/forum?id=1I7PCbOPfe*

## Abstract

While scaling Transformer-based large language models (LLMs) has demonstrated promising performance across various tasks, it also introduces redundant architectures, posing efficiency challenges for real-world deployment. Despite some recognition of redundancy in LLMs, the variability of redundancy across different architectures in transformers, such as MLP and Attention layers, is under-explored. In this work, we investigate redundancy across different Transformer modules, including blocks, MLP layers, and attention layers, through the lens of layer dropping. Surprisingly, despite the pivotal role of attention mechanisms in distinguishing Transformers from other architectures, we find that a large portion of attention layers exhibit excessively high redundancy and can be pruned without degrading performance. For example, LLaMA-3-70B achieves a 43.4% speedup with only a 1.8% drop in performance by pruning half of its attention layers. In contrast, dropping MLP layers severely impairs the model's ability to distinguish between tokens, leading to catastrophic performance degradation. Moreover, our analysis reveals that attention layer redundancy persists not only throughout training but is also evident in randomly initialized models. We attribute this redundancy to three key factors that constrain representational updates from attention layers: sparse attention patterns, over-smoothed token embeddings, and the low representational magnitude of attention outputs. Overall, our findings offer valuable insights into the internal redundancy of Transformer architectures and provide practical guidance for designing more efficient LLMs. The code is released at: `https://github.com/CASE-Lab-UMD/LLM-Drop`.

## 1 Introduction

Transformer-based large language models (LLMs) have significantly advanced AI research, achieving remarkable performance across various domains (OpenAI, 2024; Team, 2024). However, scaling these models also introduces redundant architectures, namely overarching, leading to inefficiencies that complicate their real-world deployment (Frantar et al., 2023; Sun et al., 2023), e.g., inflating deployment costs and resource demands. Although several previous works have been proposed to promote LLM efficiency via removing redundant parameters or architectures (Frantar et al., 2023; Sun et al., 2023), these approaches often employ universal techniques that overlook the unique characteristics of transformer architectures. Specifically, transformer (Vaswani, 2017) architectures are composed of multiple stacked blocks, each containing an MLP layer and an Attention layer, which serve distinct functions and exhibit corresponding different levels of redundancy (Wang et al., 2023; Shi et al., 2023) This motivates a deeper investigation into the specific redundancies within Transformers, with the goal of identifying and addressing the most critical modules.

In this work, we systematically explore the redundancy in three key Transformer components: *Block*, *MLP*, and *Attention*. Using a similarity-based metric (Gromov et al., 2024; Men et al., 2024) , we evaluate the importance of each component and progressively drop those identified as redundant. We first apply a "*Block*

---

[*]Equal contribution

*Drop*" approach but observe that removing entire blocks leads to significant performance degradation. This suggests a need for a more fine-grained strategy.

Upon further examination, we explore the separate pruning of MLP and Attention layers. Our findings reveal that while dropping MLP layers negatively affects performance, a substantial portion of Attention layers, i.e., the core of Transformer architectures which distinguish it from other mainstream architectures (e.g., RWKV (Peng et al., 2023b) and Mamba (Gu & Dao, 2024)), can be pruned without degrading the model's performance. For instance, dropping 50% of the Attention layers in Llama-3-70B (Touvron et al., 2023) results in comparable performance to the full model, indicating a high degree of redundancy in these layers.

Our work provides a unified analysis of two key architectural components in Transformers: MLP and attention layers. On the one hand, MLP layers, which perform token transformations in high-dimensional space, contribute significantly to the hidden state representations. Without them, Transformer models suffer degraded sensitivity to token distinctions, resulting in substantial performance loss. In contrast, attention layers display notable redundancy. This redundancy persists across various training stages and is even observable in randomly initialized models, indicating that it is an inherent property of the architecture. We attribute this phenomenon to three key factors in attention architectures: (1) the sparsity of attention matrices, (2) the excessively high similarity between token embeddings, and (3) the low representational magnitude of the attention branch. These factors collectively constrain the representational updates contributed by attention layers to the residual branch, thereby making certain attention layers amenable to pruning. These findings offer valuable insights into the internal redundancy of Transformer architectures and provide practical guidance for designing more efficient LLMs. In summary, our key contributions are as follows:

- Through an in-depth analysis of redundancy in three key Transformer components, we uncover a surprising level of redundancy within the *Attention*.

- We propose *Attention Drop*, a simple yet effective algorithm for removing less important attention layers in a training-free manner, significantly improving efficiency without sacrificing performance.

- Our extensive experiments demonstrate the effectiveness of dropping Attention, for instance, removing 50% of the attention layers in Llama-3-70B results in only a 1.8% performance reduction while achieving up to a 43.4% speedup.

- We further demonstrate that attention layers possess inherent properties that lead to consistently high redundancy, both at initialization and throughout the training process. This observation offers valuable insights for future architectural design.

## 2 Related Works

**Large Language Models**    Although Transformer-based Large Language Models (LLMs) have demonstrated promising performance across various tasks, their deployment costs still remain a significant challenge for practical usage (Sun et al., 2023; Lin et al., 2024; Gromov et al., 2024). Transformer (Vaswani, 2017) models consist of multiple blocks, which include Attention layers and MLP layers. Attention layers compute the contextual information between input tokens with quadratic complexity concerning the input sequence length (Li et al., 2020). KV-Cache (Pope et al., 2022) mitigates the computational issue but results in excessive memory costs (Zhang et al., 2023). MLP layers (Liu et al., 2021; Mai et al., 2022) transform each token independently, using an up-projection followed by a down-projection, and contribute most of the model parameters. Recent works have revealed that not all blocks or layers are equally important (Men et al., 2024; Chen et al., 2024), which urges us to reflect on the structured redundancy within LLMs and the potential design of more compact architectures.

**Model Compression**    LLMs can be compressed to promote their efficiency in memory and computation. Quantization (Frantar et al., 2023; Lin et al., 2024) and Pruning (Sun et al., 2023; Frantar & Alistarh, 2023) are the most widely used techniques to compress LLMs. Specifically, quantization transforms the data type into low-bit but remains potentially redundant architecture and parameters. Pruning can be categorized into unstructured pruning (Kusupati et al., 2020; Sanh et al., 2020) and structured pruning (Zhuang et al., 2020;

Kwon et al., 2020). While unstructured pruning maintains better performance than structured pruning, it cannot be effectively applied to hardware, limiting its practical usage. Our methods, Block Drop and Layer Drop, focus on removing structured modules rather than fine-grained parameters, creating hardware-friendly efficient architectures while maintaining comparable performance. Additionally, Block Drop and Layer Drop are orthogonal to quantization, and their integration with quantization significantly enhances efficiency.

## 3 Methodology

In this section, we present the methodology for identifying and removing redundant modules in LLMs. We begin by introducing a similarity-based metric to assess redundancy for layer importance. Based on the insights gained from this analysis, we develop two targeted techniques, i.e., MLP Drop and Attention Drop, to efficiently eliminate redundant components while preserving model performance.

### 3.1 Preliminaries

**Similarity-based Drop** To assess the redundancy of modules in LLMs, we employ a similarity-based metric that evaluates the importance of each module by measuring the similarity between its input and output (Gromov et al., 2024). The underlying hypothesis is that redundant modules produce outputs that are similar to their inputs, implying minimal transformation. In contrast, important modules are expected to significantly alter their inputs and thus should be preserved. The similarity between the hidden states of the input $X$ and output $Y$ of a module is quantified using cosine similarity. The importance score $S$ of the module is computed as:

$$S = 1 - \text{CosineSim}(X, Y). \tag{1}$$

Modules with higher cosine similarity exhibit lower importance scores, indicating redundancy. We identify and prune the modules with the lowest importance scores according to a predefined prunting ratio. A complete evaluation of the effectiveness of cosine similarity as the dropping metric is provided in Table 5.

**Block Drop** Transformer models are composed of stacked blocks, where each block shares a common architecture and can be viewed as a subnetwork. To reduce complexity, we first consider dropping entire blocks that are deemed unimportant.

As shown in Figure 2, Transformer blocks operate sequentially, with each block's output feeding into the next. To evaluate redundancy, we compute the similarity between the input and output of each block. For the $l$-th block, the importance score is calculated as:

$$S_B^l = 1 - \text{CosineSim}(X_B^l, Y_B^l), \tag{2}$$

where $X_B^l$ and $Y_B^l$ denote the input and output of the $l$-th block, respectively. Since the similarity scores are computed locally, we can offload irrelevant modules to save memory. By iteratively computing the importance scores for each block from shallow to deep, we can identify and drop blocks with the lowest scores, thus saving memory and computational resources.

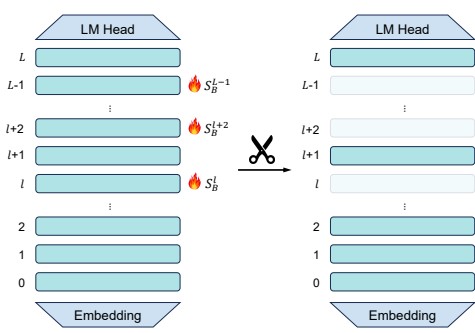

Figure 1: **Visualization of Block Drop**. where we use 🔥 to denote the blocks with high similarity scores. The dropped blocks are blurred.

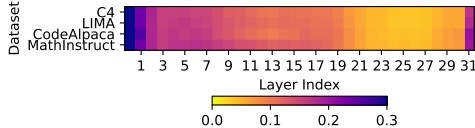

Figure 2: Importance scores of Blocks.

### 3.2 Motivation

Block Drop is an aggressive technique that risks removing essential layers, as it overlooks the internal fine-grained architectures within each block. Given a transformer block consists of both an Attention layer and an MLP layer. These layers perform distinct functions, with the Attention layer facilitating contextual

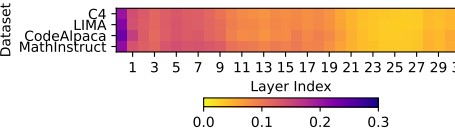 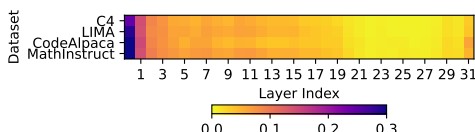

Figure 3: Importance scores for MLP (left) and Attention (right) layers, using various calibration datasets for comprehensive analysis.

information flow between tokens and the MLP layer transforming the token representations. Given their distinct roles, we assess the redundancy of each layer separately by measuring the importance scores of Attention and MLP layers individually. Specifically, we leverage multiple calibration datasets to measure the importance scores, ranging from the pretraining dataset (e.g., C4 (Raffel et al., 2020)) to instruction fine-tuning datasets (e.g., CodeAlpaca-20k (Rozière et al., 2024), MathInstruct (Yue et al., 2024) and LIMA (Zhou et al., 2024)). Figure 2 and 3 illustrate the varying trend of importance scores for Attention layers compared to MLP layers across multiple datasets. This observation motivates us to consider the varying levels of redundancy between MLP and Attention layers and to develop more fine-grained dropping techniques accordingly, namely, MLP Drop and Attention Drop.

### 3.3 Layer Drop.

**MLP Drop** As illustrated in Figure 4, each MLP layer follows a LayerNorm operation and involves a residual connection, ensuring that part of the input is preserved in the final output. Given the input $\boldsymbol{X}_M^l$ of the LayerNorm before MLP at the $l$-th Block, the output $\boldsymbol{Y}_M^l$ can be formulated as:

$$\boldsymbol{Y}_M^l = \boldsymbol{X}_M^l + \text{MLP}(\text{LayerNorm}(\boldsymbol{X}_M^l)). \tag{3}$$

Since the output $\boldsymbol{Y}_M^l$ contains both the residual and the MLP transformation, evaluating similarity based solely on the MLP's output can be misleading. To address this, we consider the MLP layer and its associated LayerNorm as a single unit and compute their importance score as follows:

$$\boldsymbol{S}_M^l = 1 - \text{CosineSim}(\boldsymbol{X}_M^l, \boldsymbol{Y}_M^l). \tag{4}$$

By treating these layers as a single entity, we ensure a more accurate measure of importance. MLP Drop removes both the unimportant MLP and associated LayerNorm layers. Further validation of this approach can be found in Appendix A.

**Attention Drop** Similarly, Attention layers also operate within a residual connection. The output of the $l$-th Attention layer is computed as:

$$\boldsymbol{Y}_A^l = \boldsymbol{X}_A^l + \text{Attention}(\text{LayerNorm}(\boldsymbol{X}_A^l)), \tag{5}$$

where $\boldsymbol{X}_A^l$ is the inputs of the corresponding LayerNorm layers and $\boldsymbol{Y}_A^l$ overall outputs that involves residual connections. Like MLP Drop, we assess both the Attention layer and its associated LayerNorm as a single unit. The importance score for the Attention layer is:

$$\boldsymbol{S}_A^l = 1 - \text{CosineSim}(\boldsymbol{X}_A^l, \boldsymbol{Y}_A^l). \tag{6}$$

Layer Drop, for both MLP and Attention layers, is performed in a one-shot manner, calculating importance scores once and removing redundant layers in a single step. This approach avoids the resource-intensive and time-consuming iterative pruning process. The effectiveness of this simple one-shot technique is evaluated in Appendix A.

## 4 Implementation Details

**Models** We utilize Llama-3 (Grattafiori et al., 2024) and Mistral (Jiang et al., 2023) as the default models, given their competitive performance and wide usage. We also evaluated the completely open-source model

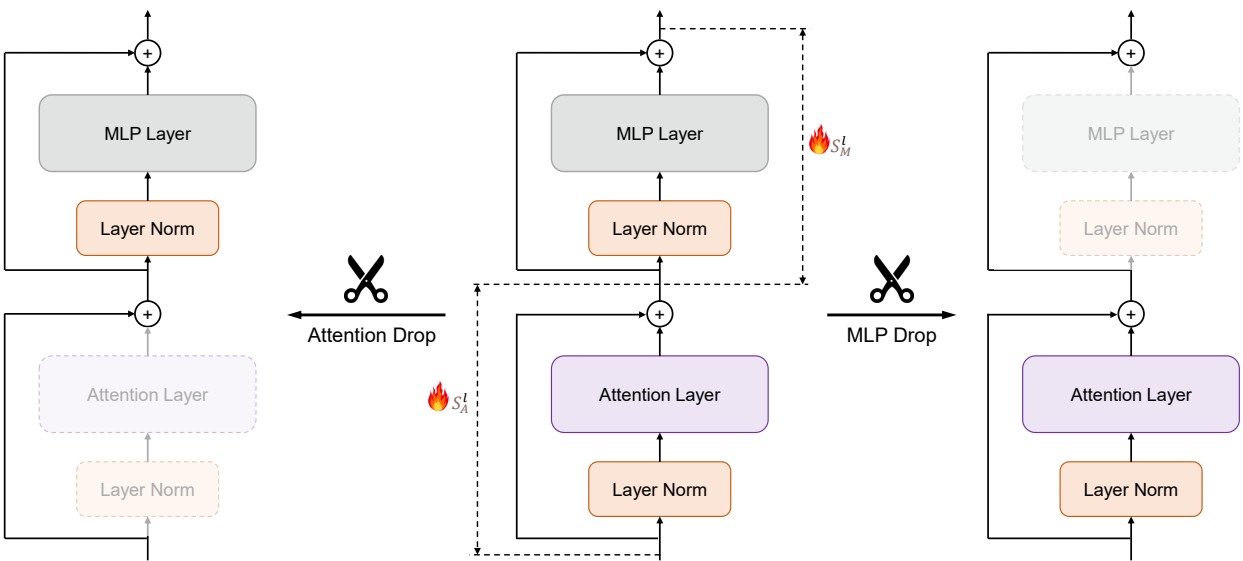

Figure 4: **Visualization of Layer Drop**, where we visualize dropping either MLP or Attention Layers. Given the residual connection, we take LayerNorm together with the corresponding layers. The dropped layers with high similarity scores 🔥 are blurred.

MAP-Neo (Zhang et al., 2024) to explore the redundancy variations in the modules of the entire pre-training phase. Additionally, we experimented with the newly released Llama-3 to verify the effectiveness of model dropping on the latest models.

**Datasets**   For the calibration dataset, we used the validation set of C4 dataset (Raffel et al., 2019), with 256 samples and an input sequence length of 2,048, following the setup in (Sun et al., 2023). The setting is well-supported by Appendix A. To evaluate model performance, we report normalized zero-shot or few-shot accuracy on the LM-harness benchmark, which includes multiple tasks: ARC-C (Clark et al., 2018), BoolQ (Clark et al., 2019), HellaSwag (Zellers et al., 2019), MMLU (Hendrycks et al., 2021), OBQA (Mihaylov et al., 2018), PIQA (Bisk et al., 2019), RTE (Wang et al., 2019), and WinoGrande (ai2, 2019). Please refer to Table 1 for detailed information. The evaluation code is based on EleutherAI LM Harness (Gao et al., 2023).

Table 1: **Experimental settings for evaluation tasks.** "Norm" refers to the normalization performed with respect to the length of the input.

| Task | Number of few-shot | Metric |
|---|---|---|
| BoolQ | 0 | Accuracy |
| RTE | 0 | Accuracy |
| OBQA | 0 | Accuracy (Norm) |
| PIQA | 0 | Accuracy (Norm) |
| MMLU | 5 | Accuracy |
| WinoGrande | 5 | Accuracy |
| GSM8K | 5 | Exact Match |
| HellaSwag | 10 | Accuracy (Norm) |
| ARC-C | 25 | Accuracy (Norm) |

## 5   Investigation of Dropping Different Target Modules

In this section, we conduct a comprehensive investigation into the effects of dropping different target modules. To quantify the trade-off between performance degradation and speedup, we introduce a new metric, i.e., **Speedup Degradation Ratio (SDR)**, defined as:

$$\gamma = \frac{\Delta\text{Avg.}}{\Delta\text{Speedup}}, \tag{7}$$

where $\Delta$Avg. represents the percentage change in average performance across the evaluated tasks, and $\Delta$Speedup denotes the corresponding percentage of speedup achieved by each method. Therefore, $\gamma$ measures the amount of performance degradation incurred for each 1% increase in speedup. A lower $\gamma$ value indicates that the model achieves speedup with minimal performance loss, making it more efficient. In contrast, a

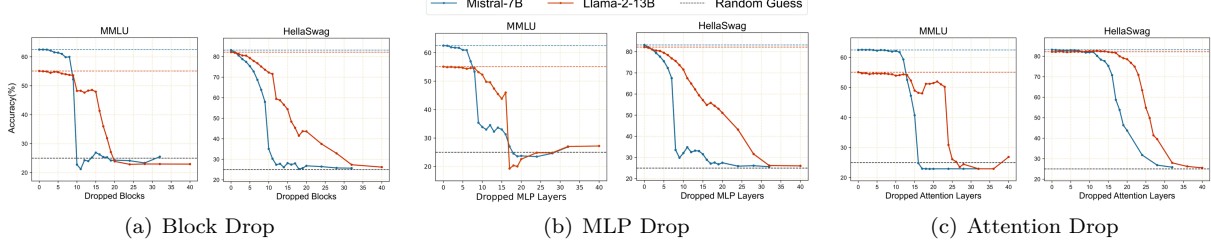

(a) Block Drop                    (b) MLP Drop                    (c) Attention Drop

Figure 5: **Performance with respect to Dropping Ratios**. The solid lines represent the impact of dropping the $n$ modules with the lowest importance scores in Mistral-7B and Llama-2-13B, and the dotted lines represent the performances of the baseline and random guessing.

higher $\gamma$ value suggests that the performance loss is substantial relative to the speedup gained, implying a less favorable trade-off.

Table 2: **Experimental results of dropping different modules**. We drop a fixed number of modules (e.g., 4 or 8) in the Mistral-7B model. The Block Drop baseline is adapted from ShortGPT (Men et al., 2024). Rows with averaged performance lower than 95% of the original performance are grayed.

| Method | ARC-C | BoolQ | HellaSwag | MMLU | OBQA | PIQA | RTE | WinoGrande | Avg. (↑) | SpeedUp (↑) | $\gamma$ (↓) |
|---|---|---|---|---|---|---|---|---|---|---|---|
| Baseline | 61.5 | 83.7 | 83.2 | 62.5 | 43.8 | 82.0 | 66.8 | 78.5 | 70.3 | 1.00× | − |
| Block-4 | 53.1 | 80.4 | 77.5 | 61.6 | 40.0 | 77.6 | 70.0 | 76.6 | 67.1 | 1.14× | 0.23 |
| Block-8 | 40.0 | 71.6 | 63.9 | 60.0 | 30.6 | 69.3 | 63.9 | 69.7 | 58.6 | 1.32× | 0.37 |
| MLP-4 | 53.2 | 80.3 | 77.7 | 61.7 | 40.0 | 77.6 | 67.5 | 77.3 | 66.9 | 1.03× | 1.13 |
| MLP-8 | 36.7 | 71.8 | 33.6 | 53.3 | 30.6 | 68.0 | 66.8 | 66.6 | 53.4 | 1.06× | 2.82 |
| Attn-4 | 61.0 | 83.5 | 82.9 | 62.5 | 44.6 | 82.0 | 64.6 | 78.0 | 69.9 | 1.10× | 0.04 |
| Attn-8 | 60.2 | 82.7 | 82.3 | 62.2 | 44.2 | 81.3 | 66.8 | 78.8 | 69.8 | 1.23× | 0.02 |
| Attn-12 | 57.2 | 76.8 | 80.2 | 59.4 | 41.8 | 79.1 | 66.1 | 77.7 | 67.3 | 1.40× | 0.08 |

Table 2 and Figure 5 summarize the results of dropping different target modules, such as Block, MLP, and Attention layers. Specifically, Table 2 shows the performance impact of dropping a fixed number of modules (e.g., 4 and 8 layers), while Figure 5 extends this analysis by evaluating a broader range of dropping ratios (0% to 100%).

**Cosine Similarity as an Effective Metric for Layer Dropping**  Building on the implementation details provided in Appendix 4, we first compare the effectiveness of various layer dropping metrics against cosine similarity. Specifically, we evaluate Reverse Order (dropping from the deepest layers upward) and Relative Magnitude (the magnitude of the layer output relative to the residual branch) metrics proposed by ShortGPT (Men et al., 2024) and apply them to the Attention Drop setting. In addition, we take the random dropping of attention layers as a baseline. All experiments were conducted

Table 3: Comparison of Dropping Metrics. The reported results reflect the average performance across HellaSwag, MMLU, OBQA, and WinoGrande.

| Metric | Attn-4 | Attn-8 | Attn-12 |
|---|---|---|---|
| Random | 61.5 | 49.6 | 39.4 |
| Reverse Order | 66.9 | 66.9 | 61.5 |
| Relative Magnitude | 67.0 | 66.8 | 62.3 |
| Cosine Similarity | **67.0** | **66.9** | **64.8** |

using the Mistral-7B model, with performance averaged over five different random seeds. As shown in Table 3, our proposed metric, Cosine Similarity, consistently outperformed the others and is therefore selected as the default metric for dropping decisions.

**Block and MLP Drop: Significant Performance Degradation with Moderate Speedup**  Block Drop and MLP Drop both lead to notable performance declines across both models, despite achieving moderate speedups. For instance, dropping 8 blocks causes a 11.7% performance drop, which is unacceptable in practice. Similarly, MLP Drop exhibits a comparable trend, with a small decline at 4 layers (3.4%,

Table 4: **Comparison with Wanda under the same parameter budget.** Even with performance-favorable unstructured sparsity, Wanda is consistently outperformed by Attention Drop.

| Method | HellaSwag | MMLU | OBQA | Winogrande | Avg. (↑) | SpeedUp (↑) |
|--------|-----------|------|------|------------|----------|-------------|
| Wanda  | 82.3 | 54.8 | 45.2 | 77.6 | 65.0 | 1.00× |
| Attn-4 | 82.0 | 54.7 | 46.2 | 77.2 | 65.0 | 1.05× |
| Wanda  | 82.4 | 54.7 | 45.8 | 77.6 | 65.1 | 1.00× |
| Attn-8 | 82.2 | 54.5 | 47.0 | 77.4 | 65.3 | 1.13× |
| Wanda   | 82.4 | 54.8 | 46.2 | 77.4 | 65.2 | 1.00× |
| Attn-12 | 82.7 | 54.4 | 48.0 | 76.6 | 65.4 | 1.20× |

Table 5: Experimental results on Llama-3-8B and Llama-3-70B.

| Method | HellaSwag | MMLU | OBQA | WinoGrande | Avg. (↑) | SpeedUp (↑) | γ (↓) |
|--------|-----------|------|------|------------|----------|-------------|-------|
| Llama-3-8B | | | | | | | |
| Baseline | 82.2 | 65.5 | 45.0 | 77.7 | 67.6 | 1.00× | – |
| Attn-8  | 81.1 | 65.1 | 45.0 | 78.4 | 67.4 | 1.16× | 0.01 |
| Attn-12 | 79.4 | 63.9 | 42.2 | 77.8 | 65.8 | 1.26× | 0.07 |
| Attn-16 | 71.2 | 38.2 | 39.4 | 72.8 | 55.4 | 1.38× | 0.32 |
| Llama-3-70B | | | | | | | |
| Baseline | 88.0 | 78.7 | 48.4 | 85.4 | 75.1 | 1.00× | – |
| Attn-32 | 87.9 | 78.6 | 48.8 | 85.3 | 75.2 | 1.35× | 0.00 |
| Attn-40 | 85.2 | 77.1 | 48.0 | 82.8 | 73.3 | 1.43× | 0.00 |
| Attn-48 | 81.2 | 73.9 | 47.4 | 81.3 | 71.0 | 1.55× | 0.07 |

$\gamma = 1.13$), but a much larger drop at 8 layers (13.5%, $\gamma = 2.82$). **These results suggest that while Block and MLP Drop provide moderate speedup, they do so at the cost of significant performance degradation, especially at higher drop ratios.**

**Attention Drop: Minimal Performance Impact with High Efficiency** Surprisingly, despite the critical role of attention layers in Transformer architectures, dropping attention layers is highly effective. Mistral-7B maintains over 95% of the original performance even after dropping 12 attention layers with a speedup of 1.40× and a low $\gamma$ of 0.08, as shown in Table 2. **These results demonstrate that attention layers are highly redundant, and their removal has minimal impact on model accuracy, making Attention Drop a highly efficient pruning strategy.**

To verify the consistency of our Attention Drop findings on larger models, we evaluate LLaMA-3-70B as shown in Table 5, which demonstrates a speedup of 1.55 and a $\gamma$ of 0.07 when dropping 40 out of 80 attention layers. **This robustness indicates that larger models can also tolerate the removal of a significant proportion of Attention layers without degrading performance.**

Attention Drop achieves substantial speedup with minimal performance degradation and outperforms other mainstream compression methods. For instance, in Table 4, we compare Attention Drop with the mainstream pruning method Wanda (Sun et al., 2023), which removes less important parameters based on activation-aware criteria. Under the same parameter budget, our methods outperform Wanda with unstructured sparsity. Additionally, Wanda contributes to fine-grained sparsity, which is not hardware-friendly and has limited practical usage.

# 6 Joint Layer Drop Further Enhances the Performance

While a significant proportion of attention layers exhibit high redundancy, our findings also show that some MLP layers have low importance. To further optimize model efficiency, we introduce Joint Layer Drop, which combines both Attention Drop and MLP Drop strategies. This approach leverages the redundancy in both attention and MLP layers to enhance the overall performance of the model.

**Methodology: Combining Attention and MLP Drop** The Joint Layer Drop method is implemented by first calculating the importance scores for both attention layers ($\boldsymbol{S}_A^l$) and MLP layers ($\boldsymbol{S}_M^l$) individually. By dropping both MLP and attention layers based on the combined set of scores, $\boldsymbol{S} = [\boldsymbol{S}_A^l, \boldsymbol{S}_M^l]$, this joint approach simultaneously removes the most redundant components from both layer types, thereby enhancing model efficiency while preserving performance.

**Superior Performance with Joint Layer Drop** As demonstrated in Figure 6, Joint Layer Drop consistently achieves better performance than either Attention Drop or MLP Drop alone. The process begins by exclusively dropping attention layers. However, as the dropping ratio increases and the more redundant attention layers are pruned, MLP layers start to become the next most redundant components. At this point, Joint Layer Drop begins to remove MLP layers, leading to further reductions in redundant layers without significant performance loss, e.g., after dropping 31 layers (Attention + MLP), Llama-2-13B still retains 90% of the performance on the MMLU task.

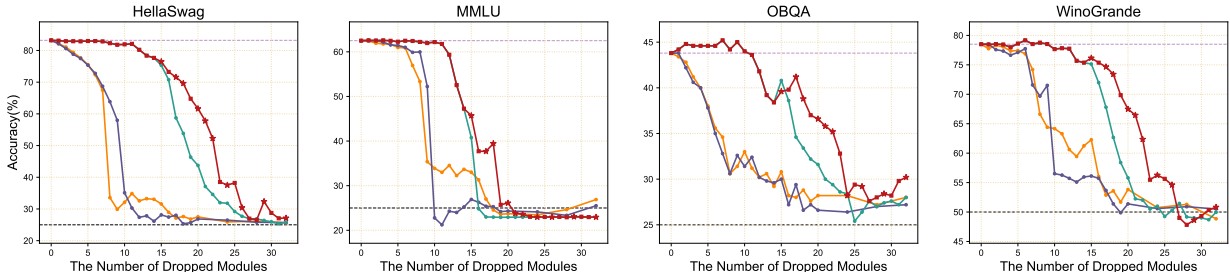

Figure 6: **Accuracy Curves of Dropping Different Target Modules**, where we consider dropping single types of modules and Joint dropping (Attn + MLP). In the line of Joint Drop, ★ represents the step where the MLP is dropped, while the ■ represents the step where the Attention is dropped.

**Comparison with other Compression Techniques**    Table 6 first compares our Joint Layer Drop with published sparse models pruned by Shortened LLaMA (Kim et al., 2024a), which estimates the importance of transformer blocks using a perplexity-based metric and performs pruning followed by retraining. Specifically, we prune the original Vicuna-13B-v1.3 model (Chiang et al., 2023) to maintain the same parameter budget. While Shortened LLaMA involves post-compression retraining, training-free Joint Drop performs better on average performance.

Table 6: Comparison with Shortened LLaMA (Kim et al., 2024a), where Joint Layer Drop achieves significantly higher speedup.

| Method | HellaSwag | MMLU | OBQA | Winogrande | Avg. (↑) | SpeedUp (↑) |
|---|---|---|---|---|---|---|
| Joint Layer Drop | 76.0 | 49.6 | 42.4 | 74.9 | 60.7 | 1.45 |
| Shortened LLaMA-PPL | 75.3 | 47.7 | 42.2 | 74.0 | 59.8 | 1.23 |
| Shortened LLaMA-Taylor | 75.8 | 47.0 | 42.4 | 74.3 | 59.9 | 1.22 |

# 7 Efficiency of Attention Drop

In this section, we evaluate the efficiency of Attention Drop in terms of both memory usage and inference speed. Specifically, we examine the reduction in memory overhead due to the key-value (KV) cache and measure the speed-up during the entire generation phase.

**KV-cache Memory Reduction** Given the auto-regressive nature of attention, where outputs are generated token by token, the KV-cache is used to store intermediate representations of input sequences. This cache helps accelerate inference by preventing redundant computations but comes with a significant memory cost, especially with longer sequence lengths or larger batch sizes. Our proposed Attention Drop method efficiently removes unimportant attention layers, reducing the corresponding KV-cache. Table 7 provides a comparison of 16-bit precision KV-cache memory usage before and after Attention Drop for

Table 7: **Comparison of KV-cache sizes before and after Attention Drop** across different models, with a sequence length of 2048. Since Llama-2-13B does not use grouped-query attention, its KV-cache costs are higher.

| Model | Batch Size | wo/Attn Drop | | w/Attn Drop | |
|---|---|---|---|---|---|
| | | Layers | KV-cache | Layers | KV-cache |
| Mistral-7B | 64 | 32 | 16GB | 20 | 10GB |
| Llama-2-13B | 32 | 40 | 52GB | 20 | 26GB |
| Llama-2-70B | 32 | 80 | 20GB | 40 | 10GB |
| Llama-3-8B | 64 | 32 | 16GB | 20 | 10GB |
| Llama-3-70B | 32 | 80 | 20GB | 40 | 10GB |

various models, where we use 8 Nvidia RTX A6000 Ada GPUs for the 70B models and 4 Nvidia RTX A6000 Ada GPUs for other smaller models. As shown, Attention Drop results in substantial memory savings across all tested models. For instance, the KV-cache decreases from 20GB to 10GB in Llama-3-70B. Note the reported results are based on resource-constrained scenarios. In resource-sufficient cases, where larger batch sizes and longer sequence lengths can be applied, the memory usage savings from Attention Drop become even more significant.

**Speed Measurement** We also evaluate the run-time speed improvements achieved through Attention Drop. The inference speed is measured throughout the entire generation process, starting from the input prompt to the generation of the final token. To ensure that the results accurately reflect the speed improvements, we follow two key principles in our setup: (1) all operations are performed on a single Nvidia RTX A6000 Ada GPU, avoiding any communication overhead caused by multi-GPU setups; and (2) we increase the batch sizes to maximize GPU utilization for each model. Specifically, for Llama-3-70B, we employ 4-bit quantization due to its large model size, while noting that Attention Drop is orthogonal to quantization shown in Appendix B. For Llama-3-8B and Mistral-7B, we use 16-bit precision. We use an input sequence of 2048 tokens and autoregressively generate an additional 2048 tokens. This setup allows us to capture the full inference process, ensuring that both the prefill and the decoding stages are included in the speed measurements.

The speed-up ratios achieved through Attention Drop are presented in Tables 2, 5, and 13. Our results show that Attention Drop provides up to 40% speed-up while retaining more than 95% of the original model's performance. Additionally, as demonstrated in Table 2, the $\gamma$ values for Attention Drop are significantly lower than those for MLP Drop and Block Drop, especially at higher speed-up ratios. This indicates that Attention Drop achieves a more efficient trade-off between speed and performance, making it a superior method for model acceleration.

# 8 Visualization and Analysis of Layer Importance

In this section, we visualize the importance scores and the corresponding dropping order of pretrained models. We then trace back through historical checkpoints to explore the dynamics of importance scores throughout the training process.

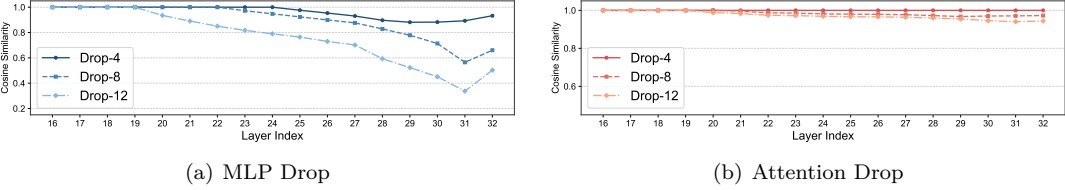

|  (a) MLP Drop | (b) Attention Drop |

Figure 7: **Layer-wise cosine similarity** between the output embeddings of the original model and models with varying numbers of dropped **(a) MLP** or **(b) Attention** layers. Since all dropped layers are located in the second half of the network, only the corresponding layers are shown for clarity.

## 8.1 Impact of Layer Drop on Embedding Representations

We have demonstrated the effectiveness of using layer-wise cosine similarity in the middle layers to guide layer dropping. We now examine how layer dropping alters the embeddings of the original model. Specifically, we extract the output embeddings of each layer from both the original and compressed models and compute the cosine similarity between them. As shown in Figure 7, dropping MLP layers leads to substantial fluctuations in the embeddings, whereas dropping attention layers maintains a high degree of similarity with the original model. The low similarity observed in the model with MLP dropping results in a significant divergence in the predicted tokens and failure to produce the correct answer, as illustrated in Figure 8. In contrast, both the original model and the model with 12 dropped attention layers are able to generate the correct answer. These

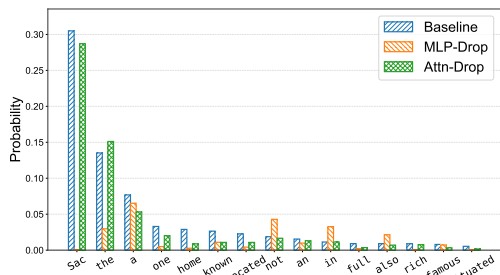

Figure 8: **Output token probability distributions** of models before and after dropping 12 layers. The input query is "The capital of California is".

findings align with the observation that MLP layers are primarily responsible for storing knowledge (Geva et al., 2021; Gromov et al., 2024).

On the other hand, since MLP layers perform token transformations in high-dimensional space, MLP dropping directly impacts the model's sensitivity to different tokens. Since the embedding of the last token depends on both itself and its preceding context, we switch the last token and monitor the resulting embedding from different forwarding passes. As shown in Figure 9, the original model exhibits low similarity across different inputs, which is consistent with the generative diversity of language models, where varying the current token leads to different subsequent tokens. In contrast, models with MLP dropping lack this capability and produce highly similar embeddings for different input tokens. This suggests that such models lose the ability to distinguish between different tokens, resulting in degraded performance. Meanwhile, models with

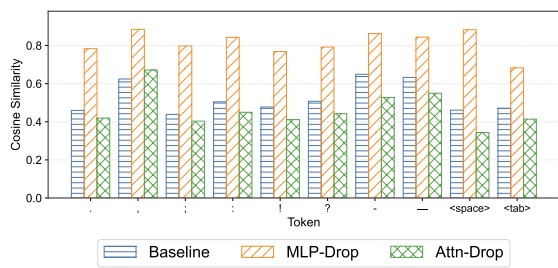

Figure 9: **Cosine similarity of the last-token embedding** before and after replacing the last token with different candidates, given the input query: "In mammals, the diaphragm is the primary muscle responsible for the act of _____".

attention dropping exhibit trends comparable to those of the original model, indicating the presence of less impactful attention layers, which we further investigate in the following sections.

## 8.2 Deeper Modules with Higher Redundancy

Based on Figure 2, 3 and 17, we observe that the deeper layers (excluding the last ones) often exhibit excessively low importance across Block, MLP, and Attention modules.

To further analyze the dropped modules, we visualize the dropped layers or blocks with different dropping ratios. Figure 10 visualizes the remaining and dropped layers/blocks as the number of dropped modules increases. Llama-2-13B and Mistral-7B exhibit similar patterns in Layer Drop and Block Drop: initially, both models tend to drop the deeper layers, followed by the shallower ones. These findings are consistent with Men et al. (2024), which suggests that deeper layers tend to be more redundant. Larger models (e.g., Llama-2-70B) also showcase a similar trend in Appendix C.

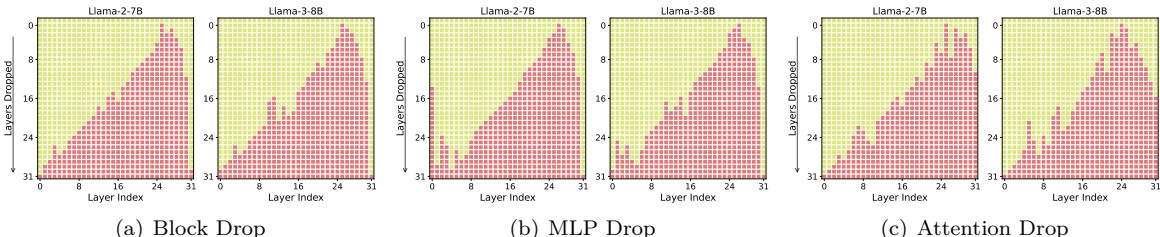

(a) Block Drop (b) MLP Drop (c) Attention Drop

Figure 10: **Visualization of Dropping Order**, where each row represents the corresponding retained and dropped layers/blocks for a specific number of dropped layers/blocks.

Previous studies on layer analysis in neural networks (Dalvi et al., 2022; Sajjad et al., 2022) have demonstrated that shallow layers are more important than deeper layers, since they primarily model local word interactions, which are critical for capturing morphology and lexical semantics. These layers provide essential input to the higher layers, and their removal can therefore be catastrophic. Building on this, we further investigate the differing behaviors of shallow and deep attention layers. Figure 11 compares the attention score matrices of a shallow layer and a deep layer.

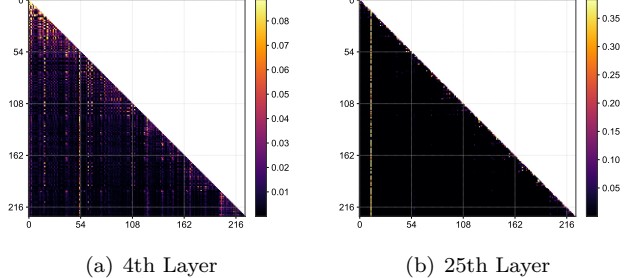

(a) 4th Layer (b) 25th Layer

Figure 11: Attention scores in the shallow and deep layer.

The shallow layer effectively captures relationships between

tokens, whereas the deep layer exhibits a sparse pattern, with attention concentrated along the diagonal and in specific columns.

## 8.3 Consistent Redundancy of Attention Layers Throughout Training

Now that the deep layers exhibit high redundancy, to investigate how such a pattern is achieved, we revisit the historical checkpoints to track the dynamic changing of layer-wise importance scores.

Specifically, we use checkpoints released by MAP-Neo-7B (Zhang et al., 2024), since it released continuous checkpoints during training stages. Figure 12 presents the importance scores of Blocks and Layers at different training stages, where Attention layers demonstrate consistently lower importance scores than MLP and Block at all training stages. While the importance scores for MLP layers and Blocks gradually increase as training progresses, the importance scores of Attention layers change much more slowly, consistently exhibiting low importance throughout the training process.

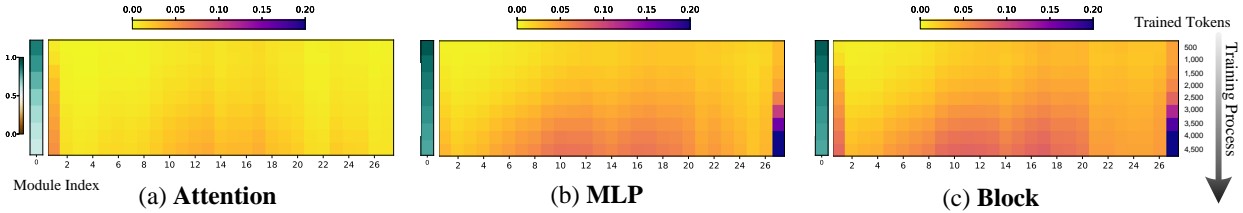

Figure 12: **Visualization of Importance Scores in Checkpoints during the Pre-training Process of MAP-Neo-7B**, where lighter areas represent low importance scores (i.e., high similarity scores). We present the entire Training Process (checkpoints for every 500B trained tokens). We independently visualize the importance scores at the module index 0, since they are significantly higher.

In addition, since randomly initialized models prior to training also exhibit low importance in certain attention layers (as shown in Figure 21), we attribute this phenomenon to intrinsic properties of the attention mechanism, which can be formulated as a weighted linear combination:

$$\boldsymbol{Y}_i = \left(\sum_{j=1}^{i} A_{ij} \boldsymbol{X}_j W_V\right) W_O = \sum_{j=1}^{i} A_{ij} \left(\boldsymbol{X}_j W_V W_O\right), \quad A = \text{softmax}\left(\frac{(\boldsymbol{X}W_Q)(\boldsymbol{X}W_K)^\top}{\sqrt{d_k}}\right), \quad (8)$$

where $\boldsymbol{Y}_i$ denotes the output of the $i$-th token, which depends on the first $i$ tokens. However, as shown by the attention sinks in Figure 11, attention disproportionately focuses on early tokens (e.g., the $t$-th token), causing the term $A_{it}\boldsymbol{X}_t$ to dominate the sum $\sum_{j=1}^{i} A_{ij}\boldsymbol{X}_j$. This sparse localized focus effectively limits the contribution of the attention branch to the residual connections.

On the other hand, Attention also has the potential to increase similarity between neighboring tokens, e.g., both the $i$-th token and $i$-th aggregate the first $i$ tokens. Through progressively aggregation in mul-

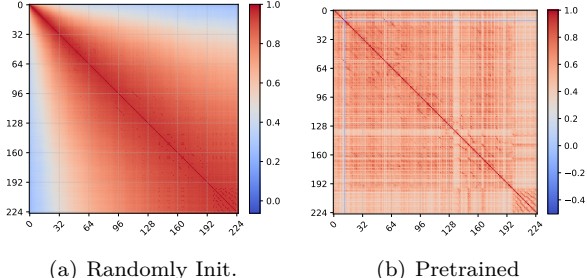

Figure 13: Token-to-token similarity in deep layers, with the complete visualization in Appendix C.

tiple layers, even distant tokens can maintain a high degree of similarity, as shown in Figure 13. Notably, the high token-to-token similarity observed in a randomly initialized model further highlights the inherent aggregation behavior of the attention mechanism. More detailed visualizations and analyses are provided in Appendix C.

Consequently, such high similarity between tokens (i.e., oversmoothed tokens (Wu et al., 2023)) limits the degree of representational updates in attention layers, since aggregating nearly identical tokens has a diminished effect on the hidden states. Moreover, due to the sparse patterns of attention weights and the low magnitude of the attention branch, as shown in Figure 14, the updates are further constrained. This limitation diminishes the functional contribution of certain attention layers, thereby reducing their relative importance within the model.

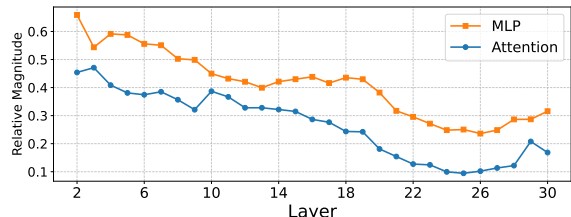

Figure 14: Relative magnitude of layer outputs compared to the residual branch.

## 9 Discussions and Conclusion

**Insights for Future Network Architecture Design**  Despite the success of scaling up LLMs, our work provides valuable insights for scaling down models to achieve more efficient architectures. One key insight is the high redundancy in attention layers, particularly in deeper layers, which suggests that future works could reduce the number of attention layers without sacrificing performance, rather than maintaining parity with MLP layers. Moreover, unlike MLP layers, attention layers exhibit consistent redundancy as training progresses. This consistency may pose a bottleneck in training large models. To address this, future research could explore replacing attention layers with alternative mechanisms or develop new training techniques that capitalize on this redundancy to further enhance language model capacity.

**Limitations**  While our proposed dropping techniques improve efficiency with minimal performance degradation, several limitations remain. An important direction for future work is to evaluate the applicability of these techniques across a broader range of architectures, such as vision transformers and vision–language models. Although our method is designed for training-free dropping without retraining, additional fine-tuning may further restore, or even improve, performance following pruning. As a preliminary study, we conduct both LoRA and full-model fine-tuning on Llama-3.2-3B using Tulu 3 (Lambert et al., 2024) as the training dataset. The results, shown in Figure 15 and detailed in Appendix B, highlight the strong potential of fine-tuning for restoring the performance of compressed models. Finally, integrating layer dropping with large-scale pretraining represents a promising direction for future research. Identifying and removing redundancy at early training stages may enable models to be trained with inherently compact architectures, improving training efficiency while supporting more fundamental optimization of model capabilities.

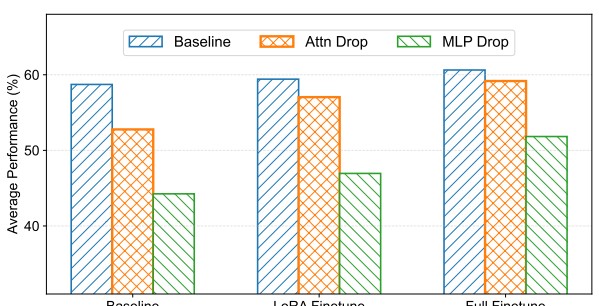

Figure 15: **Average performance of Llama-3.2-3B across post-training strategies.** Fine-tuning effectively mitigates the degradation introduced by layer dropping, with full-model fine-tuning achieving the most substantial recovery.

**Conclusion**  In this work, we systematically revisited transformer architectures by investigating the effects of dropping three types of structures: Blocks, MLP layers, and Attention layers. Our findings reveal that attention layers display significant redundancy and can be removed in large proportions without compromising performance. To build on this, we introduced Joint Layer Drop, a method that further increases both dropping ratios and performance by targeting redundant layers across both MLP and Attention layers. This study empirically demonstrates the potential for creating more compact and efficient transformer models, providing valuable insights for future network design within the NLP community. By exploring structured redundancy, we open up new avenues for designing more efficient, scalable models that maintain high performance even under resource constraints.

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

# A    Ablation Studies

**One-Shot v.s. Iterative**    One-shot and iterative approaches are the two most common methods for model compression. In the one-shot approach, importance scores are computed once, and the model is pruned in a single step. In contrast, the iterative method computes importance scores and prunes the model incrementally over multiple iterations. In Figure 16, we empirically compare Iterative Dropping and One-Shot Dropping, where in Iterative Dropping, layers are removed one by one in each iteration.

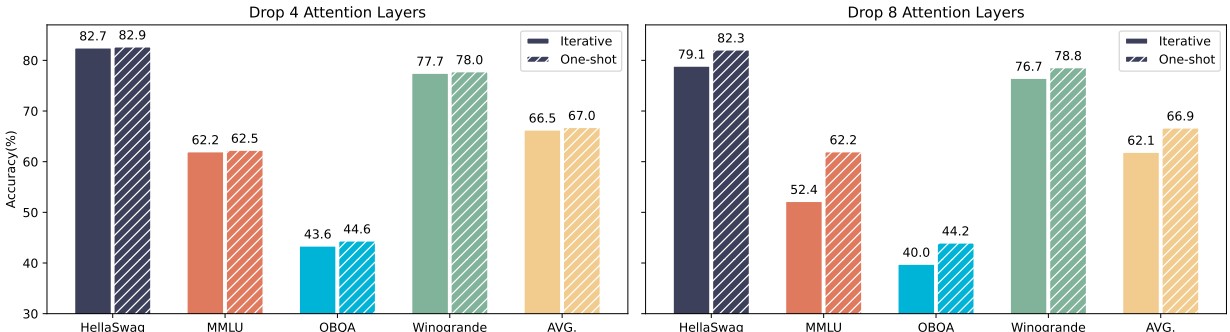

Figure 16: **Ablation Study on Dropping Strategies**, i.e., Iterative and One-Shot, where One-Shot Dropping achieves comparable performance with Iterative Dropping.

As shown in Figure 16, Iterative Dropping achieves performance that is merely comparable to One-Shot Dropping, without offering any significant enhancement. This occurs because removing a layer lowers the similarity scores of deeper layers, making them more critical. As a result, subsequent dropping steps increasingly target shallower layers, accelerating the model's collapse. Given the simplicity and efficiency, One-Shot Dropping emerges as the superior choice.

Table 8: **Ablation Study on the Residual Connection**, where we report the average performance on MMLU, WinoGrande, HellaSwag, and OpenbookQA. $n$ denotes the number of dropped modules. The notation "w/ res" indicates the involvement of the residual connection, while "w/o" indicates dropping without considering it.

|  | Attn Drop | | MLP Drop | |
|---|---|---|---|---|
| $n$ | w/o res | w/ res | w/o res | w/ res |
| 4 | 39.4 | 65.0 | 31.2 | 64.5 |
| 8 | 37.7 | 65.3 | 31.1 | 61.9 |
| 12 | 36.8 | 65.4 | 31.1 | 55.6 |
| 16 | 32.2 | 63.4 | 30.8 | 49.9 |
| 20 | 32.0 | 62.9 | 30.9 | 42.1 |

**Residual Connection**    The involvement of the residual connection ensures a more accurate estimation by accounting for the overall inputs and outputs. To explore its impact on performance, we also consider dropping modules without involving the residual connection. In this case, the importance scores are measured solely by the inputs and outputs of the Attention or MLP layers. As shown in Table 8, the involvement of the residual connection is essential for Layer Drop.

**Calibration Datasets**    Figure 3 demonstrates the robustness of the importance scores across different datasets. In Figure 17, we further verify that the importance scores remain relatively stable across various modules of Mistral-7B as the sample size increases. This stability indicates that both Block Drop and Layer Drop maintain consistency regardless of the number of samples. Consequently, we confirm that using 256 samples is sufficient for computing similarity, which serves as the standard adopted for all our experiments.

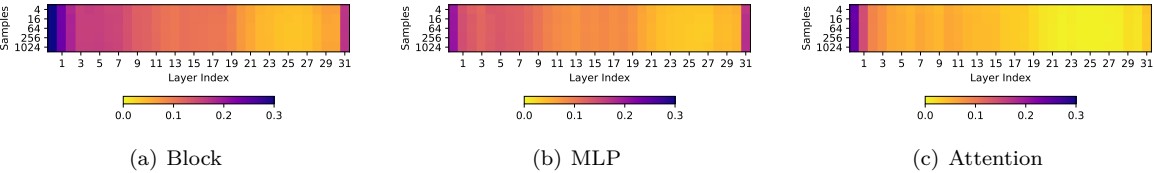

(a) Block    (b) MLP    (c) Attention

Figure 17: The impact of sample quantity on the importance scores of Block, MLP and Attention.

Table 10: Performance of various models under layer dropping.

| Method | Yi-9B | | | | | Solar-10.7B | | | | |
|---|---|---|---|---|---|---|---|---|---|---|
| | HellaSwag | MMLU | OBQA | WinoGrande | Avg. | HellaSwag | MMLU | OBQA | WinoGrande | Avg. |
| Baseline | 80.4 | 69.6 | 45.0 | 79.0 | 68.5 | 84.6 | 64.0 | 43.4 | 82.6 | 68.7 |
| MLP-4 | 75.1 | 67.2 | 38.6 | 74.0 | 63.7 | 84.0 | 63.5 | 43.6 | 81.9 | 68.3 |
| MLP-8 | 67.4 | 63.7 | 29.2 | 71.7 | 58.0 | 79.0 | 63.1 | 40.2 | 80.3 | 65.6 |
| MLP-12 | 63.4 | 60.2 | 28.6 | 69.9 | 55.6 | 73.3 | 61.5 | 36.2 | 76.6 | 61.9 |
| Attn-4 | 79.9 | 69.6 | 44.0 | 79.2 | 68.2 | 84.4 | 64.1 | 43.8 | 82.2 | 68.6 |
| Attn-8 | 79.1 | 68.2 | 42.4 | 78.2 | 67.0 | 84.4 | 64.1 | 44.0 | 83.0 | 68.9 |
| Attn-12 | 78.0 | 66.3 | 42.0 | 77.0 | 65.8 | 83.8 | 63.6 | 44.0 | 82.0 | 68.3 |
| Method | Baichuan-2-7B | | | | | Qwen3-8B | | | | |
| | HellaSwag | MMLU | OBQA | WinoGrande | Avg. | HellaSwag | MMLU | OBQA | WinoGrande | Avg. |
| Baseline | 73.8 | 54.0 | 39.6 | 70.7 | 59.5 | 76.3 | 75.0 | 38.8 | 75.1 | 65.4 |
| MLP-4 | 67.7 | 44.7 | 36.4 | 69.1 | 54.5 | 75.6 | 75.7 | 39.0 | 74.5 | 65.6 |
| MLP-8 | 54.2 | 36.8 | 33.6 | 66.6 | 47.8 | 66.5 | 70.9 | 36.8 | 69.8 | 59.2 |
| MLP-12 | 44.2 | 47.9 | 28.2 | 62.0 | 45.6 | 50.4 | 49.4 | 30.4 | 60.1 | 45.5 |
| Attn-4 | 74.1 | 54.1 | 39.8 | 71.3 | 59.8 | 79.7 | 76.5 | 41.4 | 77.0 | 68.5 |
| Attn-8 | 73.4 | 53.8 | 40.8 | 71.8 | 60.0 | 79.1 | 76.1 | 39.8 | 76.0 | 67.5 |
| Attn-12 | 65.1 | 44.9 | 37.4 | 69.7 | 54.3 | 57.0 | 57.0 | 37.2 | 62.3 | 55.9 |

**Consistent Effectiveness Across Different Models**
Tables 12 and 13 report additional results on LLaMA-2-13B and LLaMA-2-70B, which exhibit trends consistent with those observed on Mistral and Llama-3. Beyond base models, we further evaluate the instruction-tuned variant Llama-3-8B-Instruct in Table 9. In addition, we explore a broader set of architectures, including Yi (AI et al., 2025), Solar (Kim et al., 2024b), Baichuan-2 (Yang et al., 2025a),

Table 9: Experimental Results of Attention Drop on Llama-3-8B-Instruct.

| Method | HellaSwag | MMLU | OBQA | WinoGrande | Avg. (↑) | SpeedUp (↑) | γ (↓) |
|---|---|---|---|---|---|---|---|
| Baseline | 78.7 | 65.6 | 42.8 | 76.6 | 66.0 | 1.00× | – |
| Attn-4 | 78.4 | 64.9 | 44.2 | 76.6 | 66.1 | 1.07× | 0.03 |
| Attn-8 | 77.9 | 65.4 | 43.8 | 75.5 | 65.6 | 1.16× | 0.01 |
| Attn-12 | 76.3 | 63.6 | 42.8 | 75.5 | 64.5 | 1.26× | 0.07 |
| Attn-16 | 68.6 | 33.5 | 39.6 | 68.9 | 52.7 | 1.38× | 0.32 |
| Attn-20 | 42.3 | 25.1 | 31.0 | 58.2 | 39.2 | 1.52× | 0.56 |

and Qwen3 (Yang et al., 2025b), as shown in Table 10. Moreover, given that Gemma-3 combines both post-norm and pre-norm designs, we extend Attention Drop to this model to examine its robustness under different normalization strategies. Overall, these results consistently validate the effectiveness of Attention Drop across diverse model families and configurations.

Table 11: Experimental results of Attention Drop on Gemma-3 models (Peri-LN).

| Model | OBQA | PIQA | RTE | WinoGrande | BoolQ | ARC-C | HellaSwag | MMLU | Avg. |
|---|---|---|---|---|---|---|---|---|---|
| Gemma-3-1B | | | | | | | | | |
| Baseline | 38.4 | 71.9 | 68.2 | 58.6 | 75.9 | 40.1 | 55.9 | 39.9 | 56.1 |
| Attn-8 | 38.6 | 72.3 | 68.0 | 60.1 | 76.7 | 37.4 | 55.5 | 39.7 | 56.0 |
| Attn-12 | 38.0 | 71.8 | 68.6 | 60.0 | 73.0 | 38.7 | 55.0 | 38.9 | 55.5 |
| Gemma-3-4B | | | | | | | | | |
| Baseline | 46.8 | 76.4 | 74.0 | 69.1 | 84.1 | 60.4 | 75.4 | 58.3 | 68.1 |
| Attn-8 | 45.0 | 78.0 | 73.1 | 69.2 | 83.9 | 60.1 | 75.4 | 57.9 | 67.8 |
| Attn-12 | 43.4 | 75.3 | 71.4 | 67.9 | 82.4 | 59.8 | 74.6 | 56.0 | 66.4 |

**Effectiveness on Different Normalization Layouts**  Attention Drop is not restricted to Pre-LN architectures. Adapting to different normalization layouts, including Post-LN or Peri-LN (Pre + Post LayerNorm) (Kim et al., 2025), only requires shifting the measurement point of input/output features. The core logic and pruning criterion of Attention Drop remain unchanged.

While the main experiments in the paper focus on standard Pre-LN models such as LLaMA, we also evaluate Attention Drop on Gemma (Team et al., 2025), which adopts a Peri-LN structure. The results in Table 11 demonstrate consistent redundancy patterns and pruning tolerance, confirming the broader compatibility of our approach across normalization designs.

Table 12: **Experimental Results of Dropping Different Modules**, where we drop the fixed number (e.g., 4 and 8) of modules on Llama-2-13B. Here, Block, MLP, and Attn are corresponding modules. Rows with averaged performance lower than 95% of the original performance are grayed.

| | | | | | | | | Llama-2-13B | | | |
|---|---|---|---|---|---|---|---|---|---|---|---|
| **Method** | ARC-C | BoolQ | HellaSwag | MMLU | OBQA | PIQA | RTE | WinoGrande | Avg. (↑) | SpeedUp (↑) | $\gamma$ (↓) |
| Baseline | 59.9 | 80.7 | 82.2 | 55.1 | 45.6 | 80.5 | 65.0 | 77.0 | 68.2 | 1.00× | – |
| Block-4 | 54.8 | 73.3 | 80.6 | 54.8 | 45.8 | 79.1 | 60.3 | 77.5 | 65.8 | 1.11× | 0.22 |
| Block-8 | 48.0 | 56.8 | 75.3 | 53.8 | 41.2 | 75.3 | 59.9 | 75.6 | 60.7 | 1.24× | 0.31 |
| MLP-4 | 54.9 | 76.1 | 80.4 | 54.8 | 45.4 | 79.5 | 66.4 | 77.3 | 66.9 | 1.04× | 0.32 |
| MLP-8 | 49.2 | 63.4 | 75.6 | 54.5 | 42.2 | 76.0 | 59.2 | 75.1 | 61.9 | 1.08× | 0.79 |
| Attn-4 | 58.8 | 80.4 | 82.0 | 54.7 | 46.2 | 80.5 | 67.9 | 77.2 | 68.5 | 1.05× | -0.05 |
| Attn-8 | 58.2 | 80.5 | 82.2 | 54.5 | 47.0 | 80.5 | 64.3 | 77.4 | 68.1 | 1.13× | 0.01 |
| Attn-16 | 56.4 | 79.2 | 81.9 | 48.2 | 47.4 | 79.5 | 59.9 | 76.2 | 66.1 | 1.29× | 0.07 |
| Attn-20 | 53.8 | 76.9 | 78.6 | 51.5 | 44.4 | 77.6 | 59.2 | 77.1 | 64.9 | 1.40× | 0.08 |

Table 13: **Block Drop and Layer Drop on Larger Models**, where we drop a series of numbers (from 4 to 48) of modules on Llama-2-70B. Rows with averaged performance lower than 95% of the original performance are grayed.

| | | | | | | | | Llama-2-70B | | | |
|---|---|---|---|---|---|---|---|---|---|---|---|
| **Method** | ARC-C | BoolQ | HellaSwag | MMLU | OBQA | PIQA | RTE | WinoGrande | Avg. (↑) | SpeedUp (↑) | $\gamma$ (↓) |
| Baseline | 67.4 | 83.8 | 87.1 | 68.5 | 48.6 | 82.5 | 69.3 | 83.7 | 73.9 | 1.00× | – |
| Block-4 | 63.8 | 80.4 | 84.6 | 60.2 | 48.0 | 81.6 | 71.1 | 78.0 | 71.0 | 1.07× | 0.41 |
| Block-8 | 59.1 | 77.5 | 81.3 | 55.1 | 46.2 | 81.0 | 68.2 | 73.2 | 67.7 | 1.14× | 0.44 |
| Block-16 | 44.6 | 64.6 | 69.9 | 29.3 | 40.0 | 75.2 | 51.6 | 59.7 | 54.4 | 1.30× | 0.65 |
| Block-32 | 35.1 | 58.8 | 56.7 | 25.7 | 36.8 | 71.7 | 54.5 | 55.3 | 49.3 | 1.67× | 0.37 |
| MLP-4 | 65.4 | 84.0 | 86.1 | 68.7 | 46.6 | 82.9 | 68.2 | 83.4 | 73.2 | 1.04× | 0.18 |
| MLP-8 | 64.4 | 83.9 | 84.9 | 68.7 | 47.6 | 81.7 | 66.8 | 82.2 | 72.5 | 1.05× | 0.28 |
| MLP-16 | 57.5 | 53.6 | 81.6 | 69.1 | 46.0 | 79.2 | 58.8 | 81.7 | 65.9 | 1.08× | 1.00 |
| MLP-32 | 40.6 | 61.9 | 64.2 | 59.8 | 29.8 | 64.2 | 52.7 | 72.7 | 55.7 | 1.17× | 1.07 |
| Attn-4 | 67.2 | 84.0 | 87.0 | 68.6 | 48.8 | 82.5 | 69.3 | 83.3 | 73.8 | 1.06× | 0.02 |
| Attn-8 | 67.3 | 83.8 | 86.9 | 68.5 | 48.4 | 82.9 | 69.0 | 82.6 | 73.7 | 1.12× | 0.02 |
| Attn-16 | 67.8 | 83.9 | 87.2 | 68.5 | 49.0 | 83.0 | 68.2 | 82.8 | 73.8 | 1.21× | 0.00 |
| Attn-32 | 67.2 | 84.8 | 87.2 | 68.4 | 49.6 | 81.8 | 67.5 | 83.5 | 73.8 | 1.35× | 0.00 |
| Attn-40 | 63.7 | 82.8 | 84.4 | 66.2 | 46.8 | 80.1 | 66.8 | 81.3 | 71.5 | 1.48× | 0.05 |
| Attn-48 | 58.5 | 73.7 | 80.6 | 56.8 | 45.0 | 79.8 | 59.6 | 81.0 | 66.9 | 1.62× | 0.11 |

# B  Additional Experimental Results

**Magnitude Consideration in Cosine Similarity**  Our method primarily relies on cosine similarity to identify redundant layers. Although magnitude information is not explicitly included in the criterion, it is implicitly captured through the residual structure. In residual connections $y = f(x) + x$, the cosine similarity between $x$ and $y$ depends on both the direction and the magnitude of $f(x)$; a large $f(x)$ naturally reduces similarity.

Empirically, as shown in Figures 3 and 17, many attention layers exhibit cosine similarity greater than 0.95 or even 0.99. Such high similarity values can only occur when $f(x)$ is both small in magnitude and well-aligned with $x$, providing strong evidence that these layers are functionally redundant. Furthermore, Figure 14 shows

that attention outputs consistently have lower magnitude compared to MLPs, especially in deeper layers, aligning with the trend observed in layer-wise cosine similarity.

While cosine-based ranking may not be the most sophisticated criterion, it is sufficient given the small search space (only $L$ layers) and the competitive performance achieved by the pruned models. Unlike weight- or neuron-level compression, layer-level pruning is a relatively low-complexity problem, where even simple heuristics can be effective and efficient.

Table 14: **Output Cases of Model with Dropping Some Attention Layers,** where correct answers are marked in green , while incorrect answers are marked in red . Note that "####" is used to extract the correct answer.

| Question | Janet's ducks lay 16 eggs per day. She eats three for breakfast every morning and bakes muffins for her friends every day with four. She sells the remainder at the farmers' market daily for $2 per fresh duck egg. How much in dollars does she make every day at the farmers' market? |
|---|---|
| Golden Answer | Janet sells 16 - 3 - 4 = «16-3-4=9»9 duck eggs a day.\nShe makes 9 * 2 = $«9*2=18»18 every day at the farmer's market.\n #### 18 |

| Method | Output |
|---|---|
| Baseline | Janet's ducks lay 16 eggs per day.\n She eats 3 eggs for breakfast and bakes 4 eggs for muffins, so she uses 3 + 4 = «3+4=7»7 eggs per day.\nTherefore, she sells 16 - 7 = «16-7=9»9 eggs at the farmers' market.\nShe sells each egg for $2, so she makes 9 x $2 = $«9*2=18»18 per day.\n #### 18 |
| Attn-2 | Janet's ducks lay 16 eggs per day.\nShe eats 3 eggs for breakfast and bakes 4 eggs for muffins, so she uses 3 + 4 = «3+4=7»7 eggs per day.\nTherefore, she sells 16 - 7 = «16-7=9»9 eggs at the farmers' market.\nShe sells each egg for $2, so she makes 9 x $2 = $«9*2=18»18 per day.\n #### 18 |
| Attn-4 | Janet's ducks lay 16 eggs per day.\nShe eats 3 eggs for breakfast and bakes 4 eggs for muffins, so she uses a total of 3 + 4 = «3+4=7»7 eggs per day.\nTherefore, she sells 16 - 7 = «16-7=9»9 eggs at the farmers' market.\nShe sells each egg for $2, so she makes 9 x $2 = $«9*2=18»18 per day.\n #### 18 |
| Attn-6 | Janet's ducks lay 16 eggs per day.\nShe eats 3 eggs for breakfast and bakes 4 eggs for muffins, so she uses a total of 3 + 4 = «3+4=7»7 eggs per day.\nTherefore, she sells 16 - 7 = «16-7=9»9 eggs at the farmers' market.\nShe sells each egg for $2, so she makes 9 x $2 = $«9*2=18»18 per day.\n #### 18 |
| Attn-8 | Janet's ducks lay 16 eggs per day.\nShe eats 3 eggs for breakfast and bakes 4 eggs for muffins, so she uses a total of 3 + 4 = «3+4=7»7 eggs per day.\nTherefore, Janet sells 16 - 7 = «16-7=9»9 eggs at the farmers' market.\nShe sells each egg for $2, so Janet makes 9 x $2 = $«9*2=18»18 per day at the farmers' market.\n #### 18 |
| Attn-10 | The total number of eggs the ducks lay is 16.\nJane eats 3 eggs for breakfast, so there are 16 - 3 = 13 eggs left.\nJane bakes 4 eggs for muffins, so there are 13 - 4 = 9 eggs left.\nJane sells the remaining 9 eggs at $2 each, so she makes 9 * $2 = $«9*2=18} |
| Attn-12 | Dividing the total number of eggs to the number of eggs used for personal use, we get 16 - 3 - 3 = 11 eggs for selling.\nIf each egg is sold for $2, then the total amount earned from selling is 11 * 2 = $22.\nTherefore, she makes $22 every day at the farmers' market.\n #### 22 |

**Performance on Knowledge Intensive Tasks**   To assess the impact of Attention Drop on more complex technical tasks, we evaluated Llama-2-7B and Mistral-7B, and two corresponding instruction fine-tuned models, MetaMath-7B-V1.0 and MetaMath-Mistral-7B (Yu et al., 2023). The results in Figure 18 indicate that, except for Llama-2-7B-Math, which is MetaMath-7B-V1.0, all the models do not experience significant performance degradation when dropping fewer than 8 Attention layers. We speculate that this is because Llama-2-7B-Math is initialized with Llama-2-7B and undergoes instruction fine-tuning to improve its mathematical ability. Llama-2-7B-Base exhibits poor performance in mathematics, and the ability obtained solely through fine-tuning appears to be superficial. Therefore, when dropping Attention layers, Llama-2-7B-Math's ability rapidly deteriorates.

We select the first sample from the test set of GSM8K[1] as the input and display the MetaMath-Mistral-7B's raw output in Table 14. Even when dropping 10 attention layers, the model could still give the correct answer of this question, but it failed to adhere to the correct output format. However, when dropping 12 attention layers, the model is no longer able to produce the correct answer of this question. Despite this, post-training offers a potential recovery path for performance loss due to layer dropping. For instance, lightweight LoRA fine-tuning on MetaMathQA significantly boosts Mistral-7B-Math's accuracy from 11.6% to 58.2%, demonstrating its effectiveness and making it a promising avenue for further exploration.

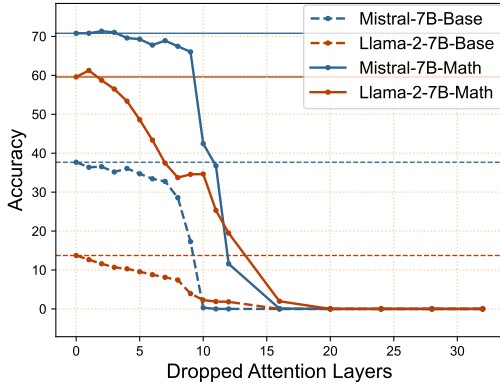

Figure 18: Accuracy Curves on GSM8K.

**Attention Drop is Orthogonal to Quantization**   Given that quantization simplifies data types and enhances efficiency in memory usage and inference speed, we integrate module dropping with quantization to verify whether the quantized models can maintain the performance achieved by Attention Drop. Specifically, we use the mainstream AWQ algorithm (Lin et al., 2024) for 4-bit quantization, following its default settings, which involve using 128 samples from the Pile dataset (Gao et al., 2020) as the calibration dataset. In Table 15, the integration of quantization still maintains the performance of Attention Drop, i.e., only less than 1% difference in average performance.

Table 15: Integration of Module Dropping and Quantization, "w/Quant" denotes quantized models.

| Method | ARC-C | HellaSwag | OBQA | WinoGrande | Avg. |
|---|---|---|---|---|---|
| Llama-2-13B | | | | | |
| Baseline | 59.9 | 82.2 | 45.6 | 77.0 | 66.2 |
| w/Quant | 59.5 | 81.7 | 45.8 | 77.1 | 66.0 |
| Attn-4 | 58.8 | 82.0 | 46.2 | 77.2 | 66.1 |
| w/Quant | 58.0 | 81.7 | 46.0 | 76.2 | 65.5 |
| Attn-8 | 58.2 | 82.2 | 47.0 | 77.4 | 66.2 |
| w/Quant | 57.7 | 81.9 | 47.0 | 77.0 | 65.9 |
| Mistral-7B | | | | | |
| Baseline | 61.5 | 83.2 | 43.8 | 78.5 | 66.8 |
| w/Quant | 61.2 | 82.5 | 42.8 | 78.0 | 66.1 |
| Attn-4 | 61.0 | 82.9 | 44.6 | 78.0 | 66.6 |
| w/Quant | 61.0 | 82.8 | 43.6 | 77.6 | 66.3 |
| Attn-8 | 60.2 | 82.3 | 44.2 | 78.8 | 66.4 |
| w/Quant | 60.1 | 82.0 | 43.8 | 77.5 | 65.9 |

Table 16: Performance on Long In-Context Task.

| Method | Sequence Length | | | | | Avg. |
|---|---|---|---|---|---|---|
| | 2k | 4k | 7k | 9k | 14k | |
| Baseline | 26 | 70 | 75 | 76 | 81 | 65.6 |
| Attn-2 | 33 | 68 | 71 | 78 | 77 | 65.4 |
| Attn-4 | 28 | 63 | 68 | 75 | 73 | 61.4 |
| Attn-6 | 24 | 63 | 65 | 72 | 69 | 58.6 |
| Attn-8 | 15 | 50 | 53 | 58 | 64 | 48.0 |
| MLP-2 | 29 | 64 | 71 | 70 | 78 | 62.4 |
| MLP-4 | 21 | 57 | 64 | 65 | 66 | 54.6 |
| MLP-6 | 9 | 41 | 48 | 47 | 48 | 38.6 |
| MLP-8 | 11 | 42 | 43 | 48 | 51 | 39.0 |

**Attention Drop on Long In-Context Task**   We evaluate the performance of Attention Drop on the long in-context benchmark. Following LongICLBench (Li et al., 2024), we present the results on BANKING77 (Casanueva et al., 2020), with the only distinction being that we sample 100 examples from the test set. BANKING77 is a banking-domain intent detection dataset comprising 77 classes. We evaluate

---

[1]https://huggingface.co/datasets/openai/gsm8k

from 1-shot/label to 5-shot/label, resulting in contextual lengths of 2k, 4k, 7k, 9k, 14k . We employ the togethercomputer/LLaMA-2-7B-32K[2] for Layer Drop, which enlarges the context window of Llama-2 to 32k using position interpolation. From the results in Table 16, we observe that Attention Drop maintains performance and outperforms MLP Drop.

Table 17: Post-training performance of LLaMA-3.2-3B under both LoRA fine-tuning and full-model fine-tuning on a 500K-sample subset of Tulu 3 (Lambert et al., 2024).

| Model | ARC-C | BoolQ | HellaSwag | MMLU | OBQA | PIQA | RTE | WinoGrande | Avg. |
|---|---|---|---|---|---|---|---|---|---|
| Baseline | 51.1 | 74.1 | 76.6 | 56.5 | 40.8 | 78.1 | 53.8 | 72.1 | 58.7 |
| w/ LoRA FT | 50.2 | 77.2 | 75.6 | 55.4 | 42.4 | 77.4 | 60.7 | 71.4 | 59.4 |
| w/ Full FT | 53.7 | 79.5 | 77.3 | 55.2 | 41.8 | 77.3 | 67.2 | 70.8 | 60.6 |
| Attn-12 | 44.4 | 74.8 | 68.2 | 31.7 | 36.2 | 72.8 | 63.2 | 69.6 | 52.8 |
| w/ LoRA FT | 46.3 | 75.5 | 71.1 | 41.4 | 41.2 | 75.0 | 69.0 | 71.2 | 57.0 |
| w/ Full FT | 50.3 | 77.5 | 73.0 | 51.9 | 39.4 | 76.8 | 73.3 | 69.3 | 59.2 |
| MLP-12 | 29.2 | 61.3 | 37.6 | 53.6 | 30.0 | 57.3 | 52.4 | 64.0 | 44.3 |
| w/ LoRA FT | 33.0 | 62.3 | 52.2 | 44.5 | 30.8 | 64.3 | 56.0 | 64.0 | 47.0 |
| w/ Full FT | 39.9 | 73.2 | 60.4 | 53.0 | 32.6 | 67.6 | 63.2 | 65.5 | 51.8 |

**Post-Training on Compressed Models**  Post-training has the potential to recover the performance degradation caused by Layer Drop. We consider two post-training scenarios: supervised fine-tuning on compressed base models and on compressed instruct models. For the former, we compress Llama-3.2-3B and perform both full-model fine-tuning and LoRA fine-tuning on Tulu 3 (Lambert et al., 2024), subsampling up to 500K training examples from the mixture. Table 17 shows that the compressed base model still provides a strong initialization for subsequent fine-tuning. We further apply LoRA fine-tuning to the compressed instruct models for one epoch, as reported in Tables 18 and 19. After lightweight fine-tuning, the compressed models demonstrate substantial performance recovery. We anticipate that continual pretraining or full-model fine-tuning could further unlock their potential.

| Method | HellaSwag | MMLU | OBQA | WinoGrande | Avg. |
|---|---|---|---|---|---|
| Baseline | 82.2 | 65.5 | 45.0 | 77.7 | 67.6 |
| + LoRA | 81.6 | 65.9 | 46.6 | 77.9 | **68.0** |
| Attn-12 | 79.4 | 63.9 | 42.2 | 77.8 | 65.8 |
| + LoRA | **81.8** | **64.9** | **45.4** | **77.8** | **67.5** |
| Attn-16 | 71.2 | 38.2 | 39.4 | 72.8 | 55.4 |
| + LoRA | **78.3** | **53.3** | **43.6** | **75.7** | **62.7** |
| Attn-20 | 42.2 | 23.0 | 30.6 | 58.7 | 38.6 |
| + LoRA | **72.8** | **31.9** | **43.4** | **69.2** | **54.3** |

Table 18: Post-finetuning of Llama-3-8B on Alpaca-GPT4 Peng et al. (2023a).

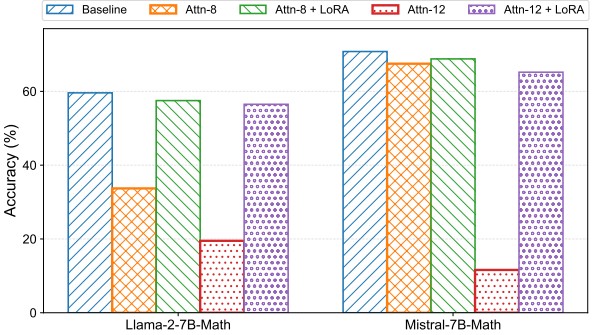

Figure 19: Post-Finetuning of Llama-2-7B-Math and Mistral-7B-Math on MetaMathQA Yu et al. (2023).

## C  Additional Visualization

**Dropping Order on Larger Models**  We present the dropping order of Block Drop and Layer Drop for the 70B Llama models in Figure 20. Similar to smaller models, larger models also tend to drop deeper layers first. While the dropping order of Blocks differs between Llama-2-70B and Llama-3-70B, we believe this is attributed to different training techniques, e.g., different numbers of training tokens.

---

[2]https://huggingface.co/togethercomputer/LLaMA-2-7B-32K

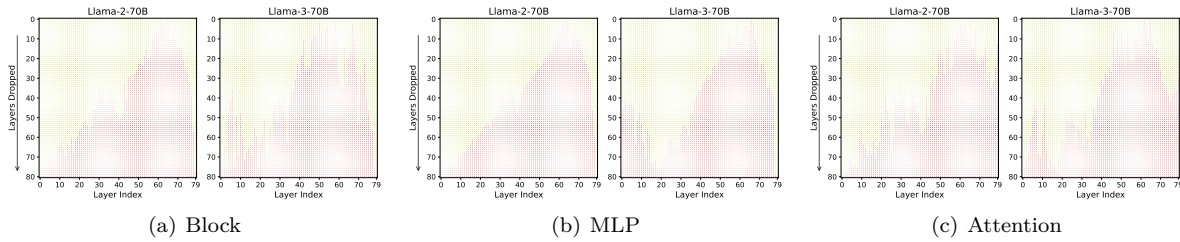

Figure 20: **Visualization of Dropping Order for Block Drop and Layer Drop on Larger Models**, i.e., Llama-2-70B and Llama-3-70B.

To provide a comprehensive view of the attention patterns and pairwise token similarity, we present the attention matrices of all attention heads in Figures 22 and 23, along with the token-wise similarity across all layers in Figures 27, 28, and 26.

**Randomly Initialized Model**   Before training, the model has not yet learned how different tokens contribute to lexical or semantic features. In addition, since the model parameters are randomly initialized, attention behaves approximately as an averaging mechanism, i.e., $\mathbf{y}_i = \frac{1}{i} \sum_{j=1}^{i} \mathbf{x}_j W_V W_O$. Noting that attention layers are deployed across stacked blocks, token embeddings are progressively aggregated along the depth of the model. Depending on the magnitude of $W_V W_O$, two scenarios arise: (1) with high magnitude, token embeddings aggregate rapidly, leading to high token-to-token similarity (Figure 27); (2) with low magnitude, limited information flow between tokens results in low similarity among token embeddings (Figure 28). Both scenarios demonstrate the

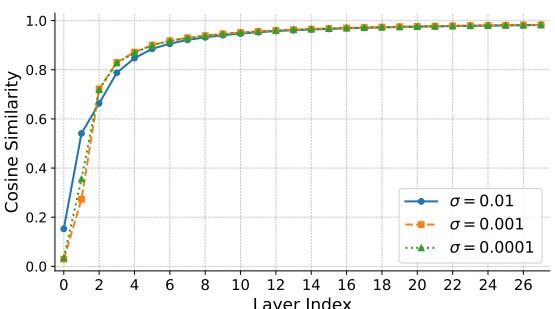

Figure 21: Layer-wise input–output cosine similarity on randomly initialized models with varying values of standard deviations $\sigma$.

redundancy of attention layers: in the first case, aggregating highly similar tokens provides minimal additional information; in the second, the low magnitude of the attention branch restricts feature updates. Together, these findings reinforce the inherent redundancy of the attention mechanism. As a result, deeper attention layers tend to exhibit excessively high input–output cosine similarity as shown in Figure 21, suggesting potential redundancy.

**Pretrained Model**   With sufficient training, the model learns to identify both lexical and semantic features, with attention sinks emerging at specific tokens (Xiao et al., 2024), which contribute significantly to overall model performance. Moreover, attention sinks in early tokens create an impactful flow of information to later tokens, such that both early and later tokens incorporate the representations of these sinked tokens. This phenomenon enhances token similarity between distant tokens. Despite this, the sparse attention pattern introduces redundancy, and the resulting high token similarity at certain depths eliminates the need for some attention operations.

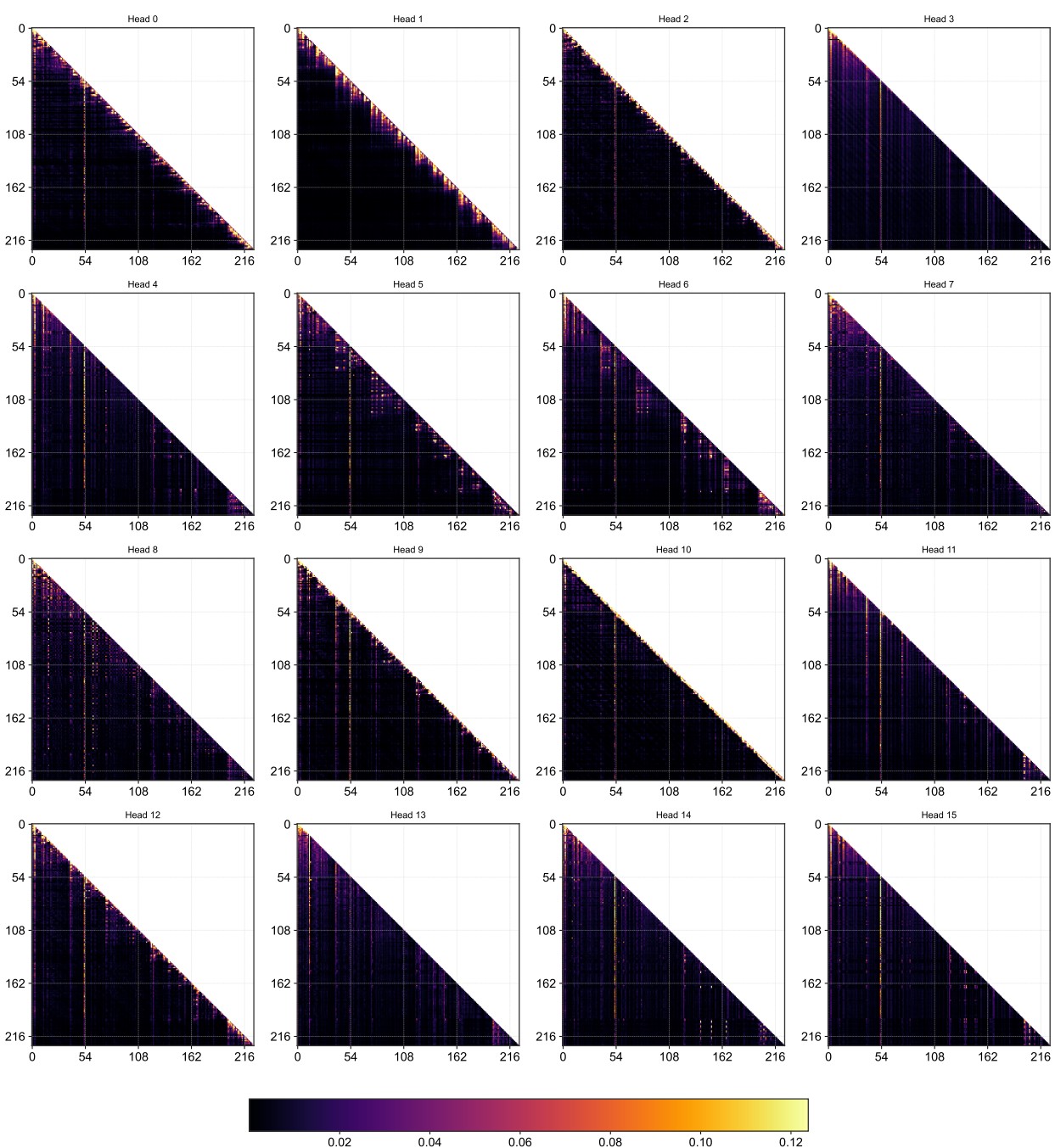

Figure 22: Attention map at the 4-th layer.

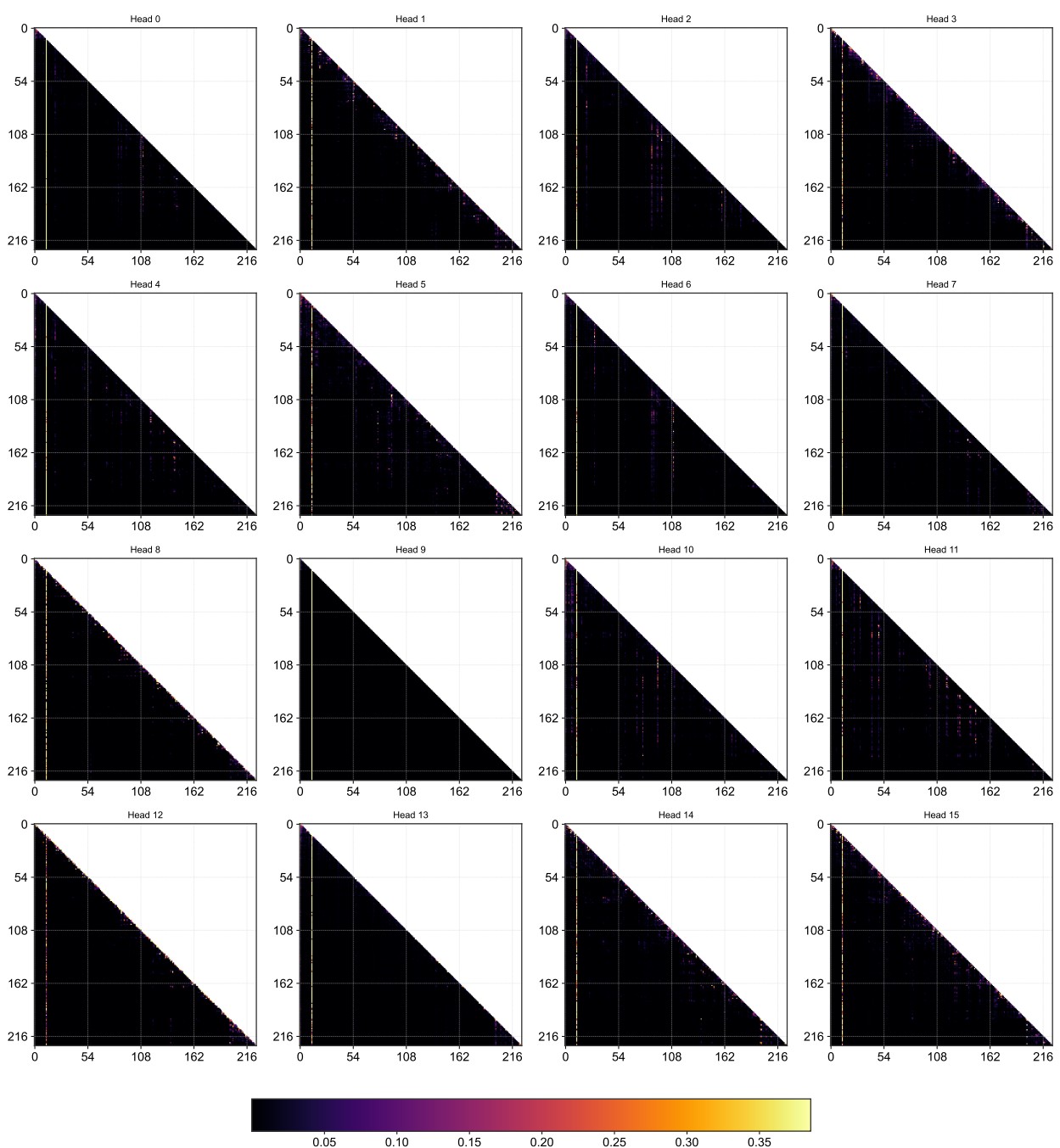

Figure 23: Attention map at the 24-th layer.

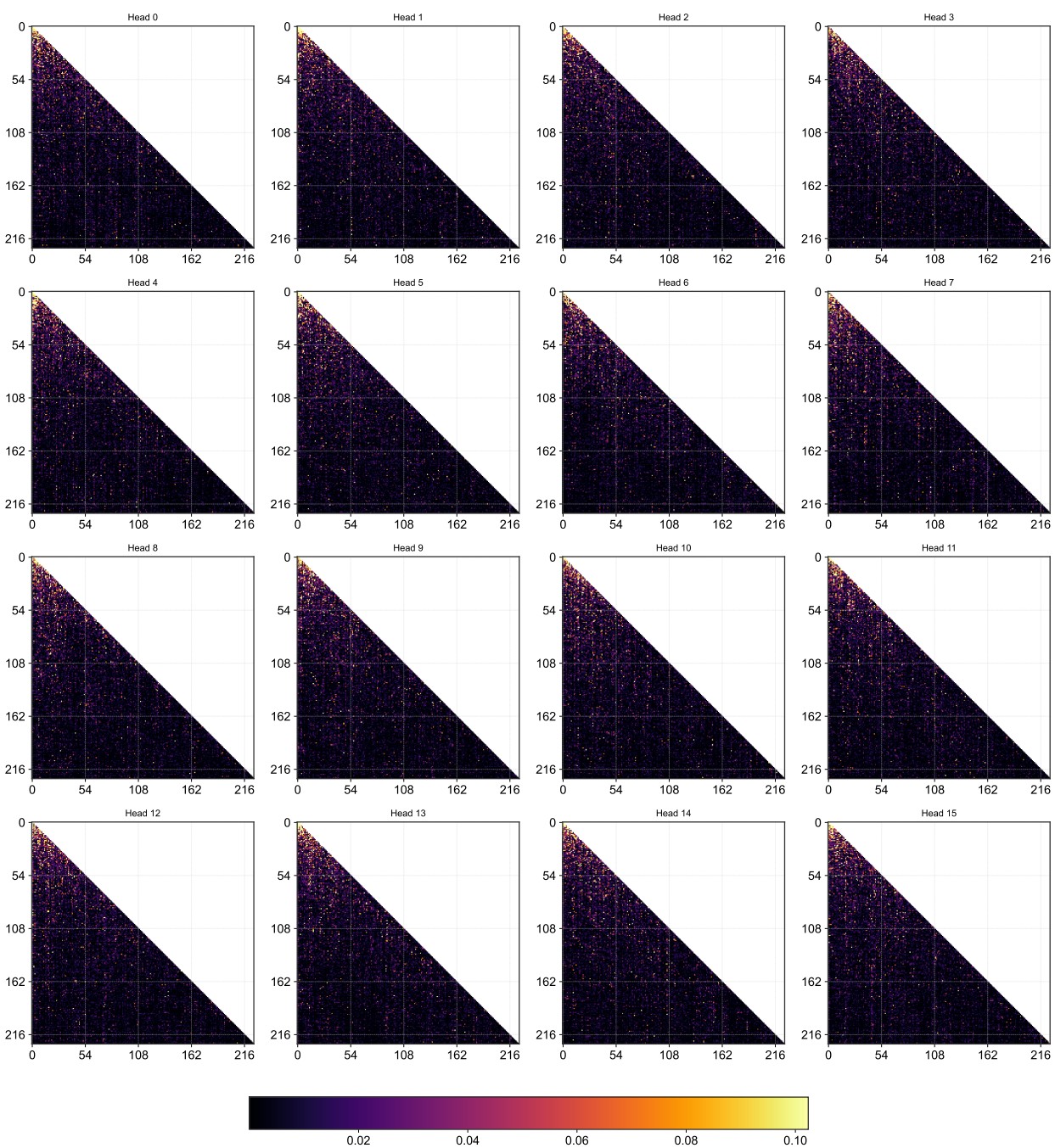

Figure 24: Attention map for randomly initialized map-neo at the 4-th layer.

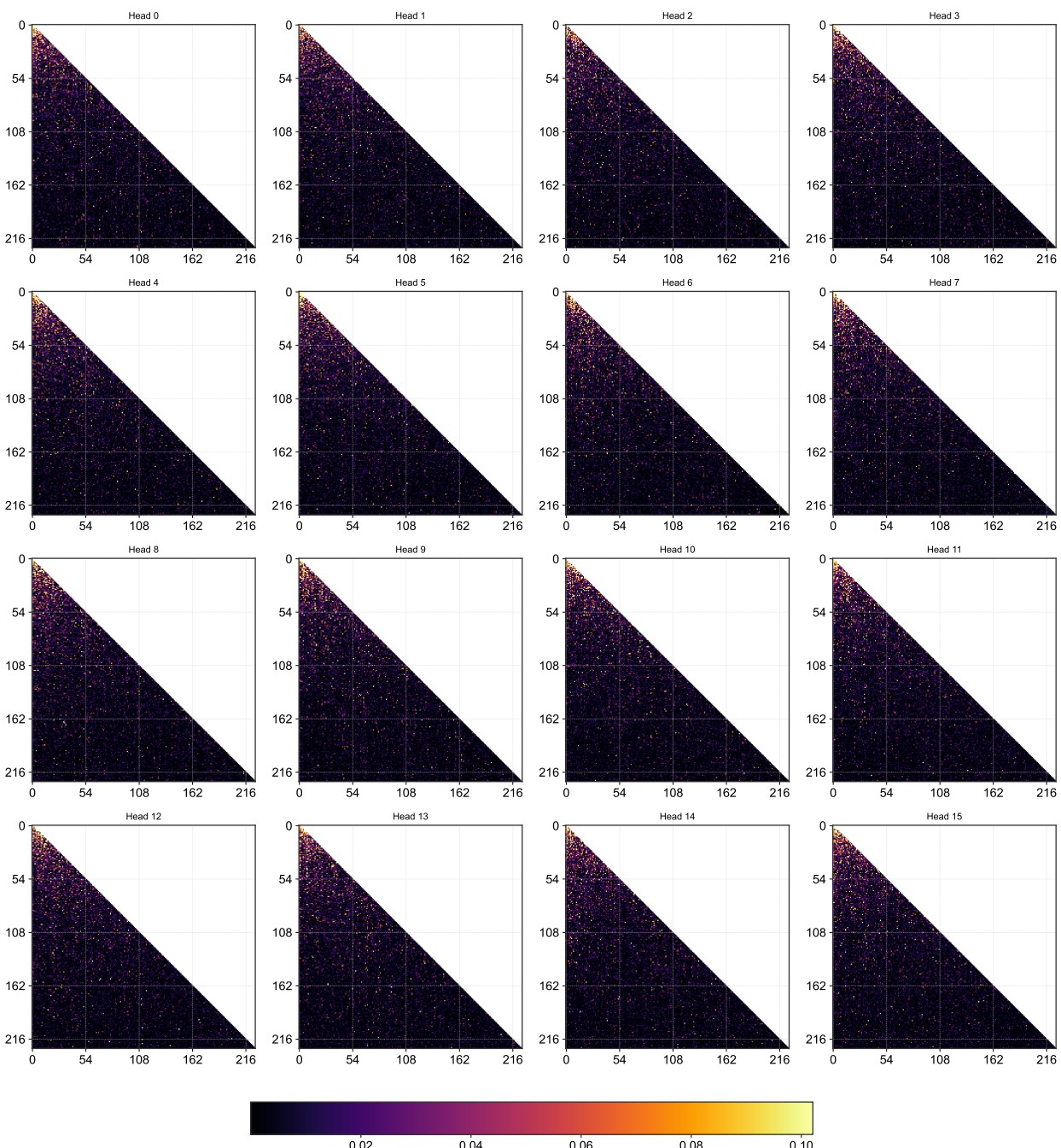

Figure 25: Attention map for randomly initialized map-neo at the 24-th layer.

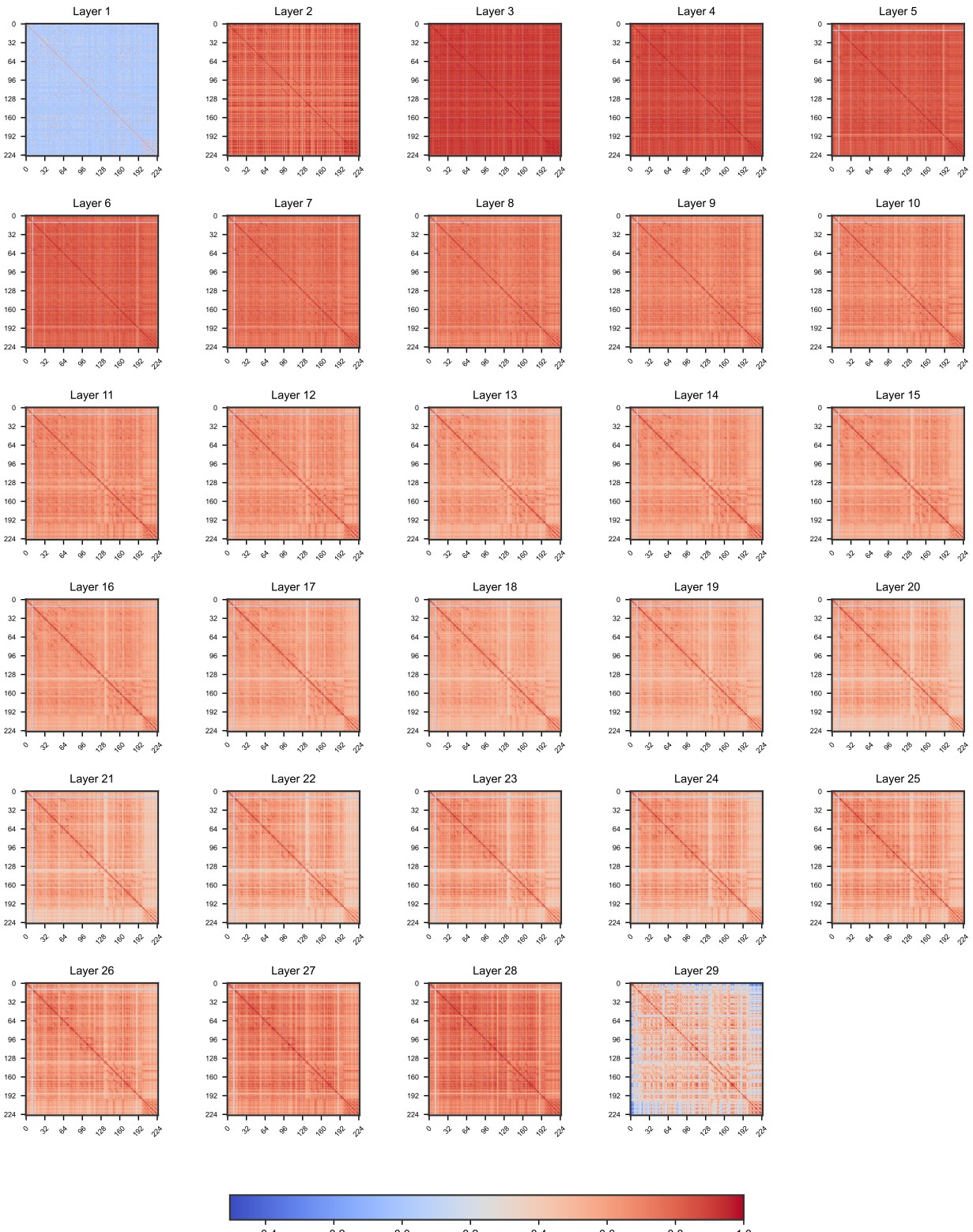

Figure 26: Token-to-token similarity in trained MAP-Neo-7B.

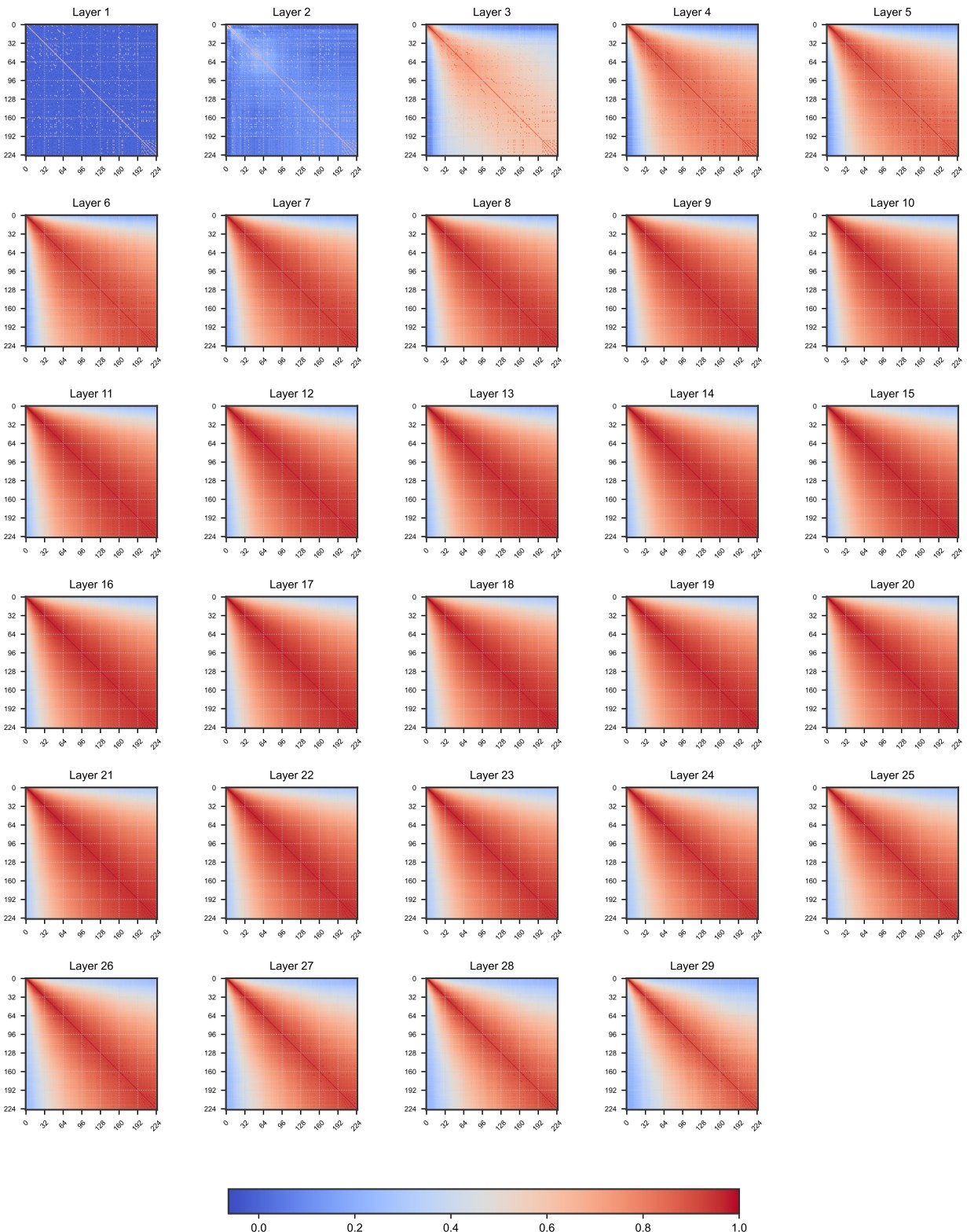

Figure 27: Token-to-token similarity in randomly initialized MAP-Neo-7B, where the parameters of the attention layers are initialized with a high standard deviation of 0.02.

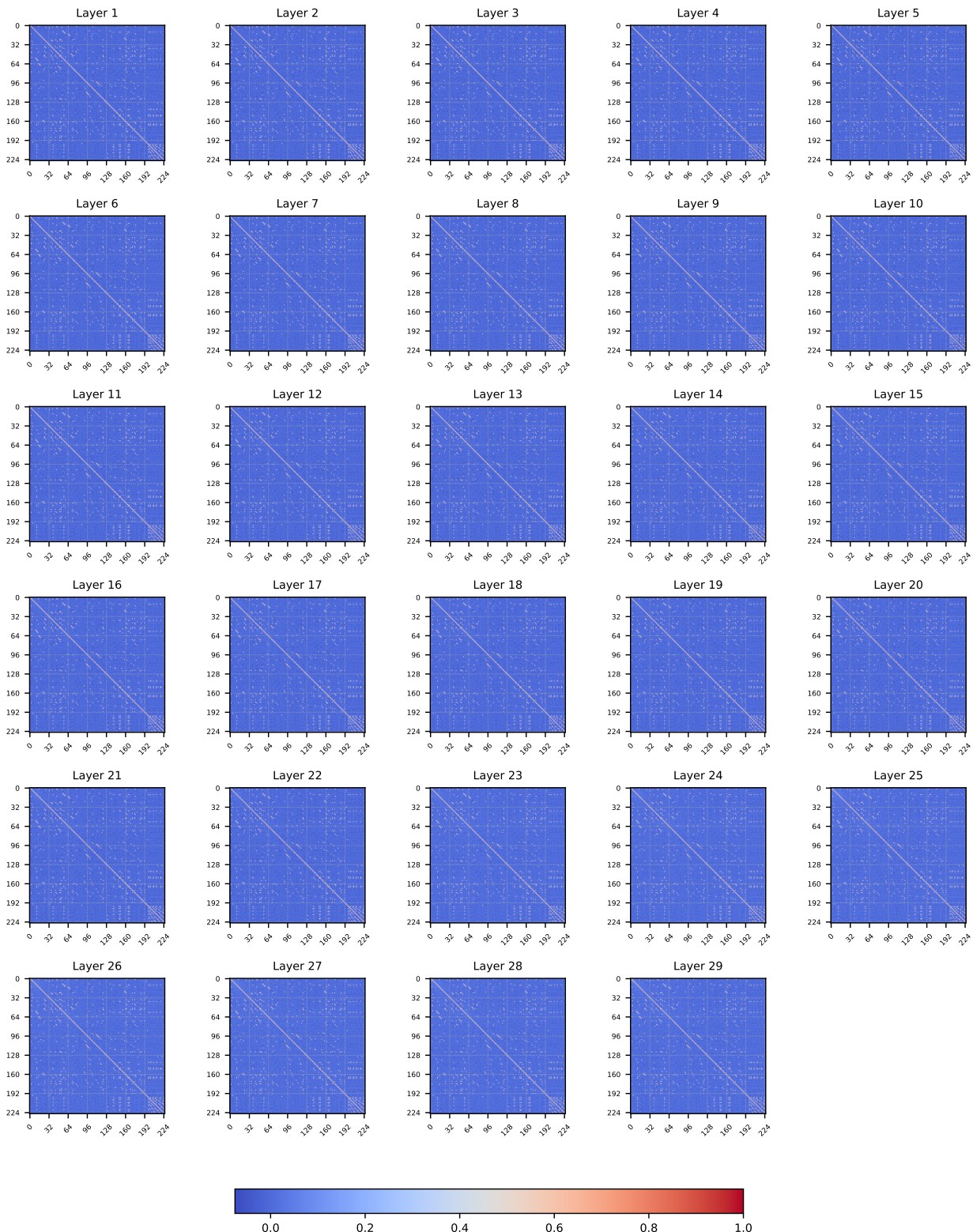

Figure 28: Token-to-token similarity in randomly initialized MAP-Neo-7B, where the parameters of the attention layers are initialized with a low standard deviation of 0.0002.

