# OpenReview forum: "Uncovering the Redundancy in Transformers via a Unified Study of Layer Dropping"
_TMLR — Accepted by TMLR_

### Review · Reviewer_bdwR · 2025-11-12

**Summary Of Contributions:**

The pruning of LLM is vital for the community because: 1). the acceleration of inference and training of LLM. 2). Further understanding of the intrinsic properties of LLM. The paper studies the pruning of LLM by carefully designed experiments and observations. Despite that the metric inherits cosine similarity, key factors (e.g., why attention layers are less important than MLP layers) revealed by the paper are impressive. Overall, it is a good and valuable paper.

**Audience:**

Yes

**Audience Explanation:**

Three summarized reasons for "pruning more attention layers" are presented in the abstract, i.e., sparse attention layers, over-smoothed token embeddings, and low representational magnitude of attention outputs. "low-representational magnitude of attention outputs" needs to be proved by more evidences. Deep attention layers indeed show sparse attention scores (Figure 10) and over-smoother token embeddings (Figure 12).

**Claims And Evidence:**

Yes

**Claims Explanation:**

The claims in the abstract are **partially** supported by the experiments, e.g., "We attribute this redundancy to three key factors that constrain representational updates from attention layers: sparse attention patterns, over-smoothed token embeddings, and the low representational magnitude of attention outputs". Please refer to the section "Explain your answer above".

**Requested Changes:**

I have the following concerns:

1. About the metric of pruning, is the cosine similarity enough? The input of a module (e.g., MLP layer, Attention layer) is similar to the output of the module under the cosine metric, where the method regards the module as the module to be pruned. That may have two flaws: 1). The cosine similarity cannot measure the high-level semantic information. 2). Is removing the module that outputs results, which do not shift from the input by a large margin, feasible? Many previous methods utilize the gradients as the metric. Do the authors consider hybrid pruning metric?

2. About the phenomenon that deep attention layers are less important and informative, that is a very valuable observation for the community.

    1). Maybe, more models can be tested for the observation, e.g., Qwen3-8B. If authors have enough GPUs, more experiments to verify the relation between the depth of the layer and the importance of the layer are expected.

    2). More discussions about less important properties of deep layers are expected. Is that related to gradient vanish in the training?

---

> ### Author Response · Authors · 2025-12-17
> **Response to Reviewer bdwR - Part I**
>
> We thank the reviewer for the thoughtful comments and provide detailed responses below.
>
> ---
>
> ## 1. Effectiveness of Cosine Similarity in Layer Dropping
>
> The current cosine similarity metric has already demonstrated strong effectiveness across various tasks for attention dropping, and prior works have also shown its effectiveness when dropping a small number of layers [1,2].
>
> One key reason why simple metrics are effective is that Block and Layer Drop operate in a low-complexity search space over the \(L\) layers. This is substantially simpler than traditional weight pruning, whose search space is on the order of
> \(\text{(number of layers × number of parameters per layer)}\),
> which is millions of times larger than that of layer dropping.
>
> As a result, layer dropping does not require sophisticated metrics. We find that simple heuristics already yield near-optimal pruning with minimal performance degradation, suggesting that more complex optimization provides limited additional benefit.
>
> Although cosine similarity does not directly model semantic meaning, it is still related to semantics, as it captures how the embedding of each token is transformed throughout the sequence.
>
> Regarding output shifts, we provide visualizations in Figures 6 and 7. Figure 6 reports layer-wise embedding similarity between the compressed model and the baseline, showing that even after dropping multiple attention layers, the model maintains high similarity to the baseline. Figure 7 visualizes the output token distributions, where dropping MLP layers causes substantial shifts, while dropping attention layers preserves a consistent distribution. These results demonstrate that cosine similarity effectively captures embedding-level and output-level changes before and after layer dropping.
>
> ---
>
> ## 2. Comparison with Other Metrics
>
> Thank you for the suggestion. We additionally consider more sophisticated metrics, including gradient-based and perplexity-based approaches.
>
> We adopt LLaMA-3-8B, as reported in Table 3, and strictly follow the same experimental settings. Among the compared approaches, the cosine-similarity-based method corresponds to our proposed approach, while the gradient-based and PPL-based methods are newly introduced baselines.
>
> For the gradient-based method, we compute gradients on the calibration datasets, average the parameter gradients within each layer, and use the resulting layer-wise gradient magnitude as the dropping metric.
>
> The PPL-based method evaluates the model’s sensitivity to layer removal by measuring performance changes on a calibration dataset. Specifically, we independently drop each attention layer and measure the resulting perplexity. The attention layers are then ranked according to the increase in perplexity from low to high, and this ranking is used to determine the layer dropping priority.
>
> | Drop_n | Metric            | HellaSwag | MMLU | OBQA | Winogrande | Avg. |
> |-------:|-------------------|-----------|------|------|------------|------|
> | 2      | Gradient-based    | 81.7 | 65.3 | 44.6 | 77.5 | 65.4 |
> |        | PPL-based         | 81.7 | 65.2 | 44.6 | 78.1 | 65.5 |
> |        | **Ours (Cosine)** | **82.1** | **65.2** | **45.0** | **78.3** | **67.7** |
> | 4      | Gradient-based    | 81.6 | 65.0 | 44.2 | 78.1 | 65.4 |
> |        | PPL-based         | 82.0 | 65.1 | 45.6 | 78.1 | 65.9 |
> |        | **Ours (Cosine)** | **81.6** | **65.1** | **44.8** | **78.2** | **67.4** |
> | 6      | Gradient-based    | 81.2 | 64.8 | 44.2 | 77.7 | 65.0 |
> |        | PPL-based         | 81.5 | 65.2 | 44.6 | 77.7 | 65.2 |
> |        | **Ours (Cosine)** | **81.8** | **65.2** | **45.4** | **78.8** | **67.8** |
> | 8      | Gradient-based    | 77.7 | 59.5 | 46.0 | 75.1 | 62.7 |
> |        | PPL-based         | 81.0 | 65.2 | 44.4 | 77.7 | 64.9 |
> |        | **Ours (Cosine)** | **81.1** | **65.1** | **45.0** | **78.4** | **67.4** |
> | 10     | Gradient-based    | 76.9 | 59.6 | 44.8 | 74.9 | 62.1 |
> |        | PPL-based         | 78.2 | 59.6 | 43.6 | 74.1 | 61.9 |
> |        | **Ours (Cosine)** | **80.5** | **64.3** | **44.8** | **78.4** | **67.0** |
> | 12     | Gradient-based    | 69.9 | 55.8 | 44.2 | 72.5 | 59.0 |
> |        | PPL-based         | 74.5 | 56.2 | 40.8 | 71.5 | 58.6 |
> |        | **Ours (Cosine)** | **79.4** | **63.9** | **42.2** | **77.8** | **65.8** |
>
> These results show that cosine similarity performs competitively with more complex metrics, indicating that the potential improvement gained from modifying the dropping metric is inherently limited. In addition, our method does not require gradient computation, and cosine similarity can be computed layer by layer, making it substantially more efficient.
>
> We believe further improvements may come from integrating Layer Drop into the training process, such as progressively dropping and retraining layers.

---

> ### Author Response · Authors · 2025-12-17
> **Response to Reviewer bdwR - Part II**
>
> ## 3. Generalization to More Models (Including Qwen3)
>
> We agree that evaluating a broader set of models strengthens the evidence for generalizability. We apply Attention Drop to a diverse set of models, including Gemma, Yi, Solar, Baichuan, and Qwen3, covering different architectures and normalization schemes. Across all cases, we observe consistent pruning robustness and competitive performance.
>
> ### Gemma Results
>
> | Model          | OBQA | PIQA | RTE | WinoGrande | BoolQ | ARC-C | HellaSwag | MMLU | Avg. |
> |----------------|------|------|-----|------------|-------|--------|------------|------|------|
> | Gemma-3-1B     | 38.4 | 71.9 | 68.2 | 58.6 | 75.9 | 40.1 | 55.9 | 39.9 | 56.11 |
> | Attn-8 (1B)    | 38.6 | 72.3 | 68.0 | 60.1 | 76.7 | 37.4 | 55.5 | 39.7 | 56.04 |
> | Attn-12 (1B)   | 38.0 | 71.8 | 68.6 | 60.0 | 73.0 | 38.7 | 55.0 | 38.9 | 55.50 |
> | Gemma-3-4B     | 46.8 | 76.4 | 74.0 | 69.1 | 84.1 | 60.4 | 75.4 | 58.3 | 68.06 |
> | Attn-8 (4B)    | 45.0 | 78.0 | 73.1 | 69.2 | 83.9 | 60.1 | 75.4 | 57.9 | 67.83 |
> | Attn-12 (4B)   | 43.4 | 75.3 | 71.4 | 67.9 | 82.4 | 59.8 | 74.6 | 56.0 | 66.35 |
>
> ---
>
> ### Yi-9B Results
>
> | Method    | Drop_n | HellaSwag | MMLU | OBQA | Winogrande | Avg. |
> |-----------|--------|-----------|------|------|------------|------|
> | Baseline  | –      | 80.4 | 69.6 | 45.0 | 79.0 | 68.5 |
> | MLP Drop  | 4      | 75.1 | 67.2 | 38.6 | 74.0 | 63.7 |
> |           | 8      | 67.4 | 63.7 | 29.2 | 71.7 | 58.0 |
> |           | 12     | 63.4 | 60.2 | 28.6 | 69.9 | 55.6 |
> | Attn Drop | 4      | 79.9 | 69.6 | 44.0 | 79.2 | 68.2 |
> |           | 8      | 79.1 | 68.2 | 42.4 | 78.2 | 67.0 |
> |           | 12     | 78.0 | 66.3 | 42.0 | 77.0 | 65.8 |
>
> ---
>
> ### Solar-10.7B Results
>
> | Method    | Drop_n | HellaSwag | MMLU | OBQA | Winogrande | Avg. |
> |-----------|--------|-----------|------|------|------------|------|
> | Baseline  | –      | 84.6 | 64.0 | 43.4 | 82.6 | 68.7 |
> | MLP Drop  | 4      | 84.0 | 63.5 | 43.6 | 81.9 | 68.3 |
> |           | 8      | 79.0 | 63.1 | 40.2 | 80.3 | 65.6 |
> |           | 12     | 73.3 | 61.5 | 36.2 | 76.6 | 61.9 |
> |           | 16     | 67.5 | 57.4 | 33.8 | 72.5 | 57.8 |
> | Attn Drop | 4      | 84.4 | 64.1 | 43.8 | 82.2 | 68.6 |
> |           | 8      | 84.4 | 64.1 | 44.0 | 83.0 | 68.9 |
> |           | 12     | 83.8 | 63.6 | 44.0 | 82.0 | 68.3 |
> |           | 16     | 82.3 | 63.5 | 43.2 | 81.5 | 67.6 |
>
> ---
>
> ### Baichuan-2-7B Results
>
> | Method    | Drop_n | HellaSwag | MMLU | OBQA | Winogrande | Avg. |
> |-----------|--------|-----------|------|------|------------|------|
> | Baseline  | –      | 73.8 | 54.0 | 39.6 | 70.7 | 59.5 |
> | MLP Drop  | 4      | 67.7 | 44.7 | 36.4 | 69.1 | 54.5 |
> |           | 8      | 54.2 | 36.8 | 33.6 | 66.6 | 47.8 |
> |           | 12     | 44.2 | 47.9 | 28.2 | 62.0 | 45.6 |
> | Attn Drop | 4      | 74.1 | 54.1 | 39.8 | 71.3 | 59.8 |
> |           | 8      | 73.4 | 53.8 | 40.8 | 71.8 | 60.0 |
> |           | 12     | 65.1 | 44.9 | 37.4 | 69.7 | 54.3 |
>
> ---
>
> ### Qwen3 Results
>
> | Method    | Drop_n | HellaSwag | MMLU | OBQA | Winogrande | Avg. |
> |-----------|--------|-----------|------|------|------------|------|
> | Block Drop | 2 | 76.3 | 75.0 | 38.8 | 75.1 | 65.4 |
> |           | 4 | 71.4 | 75.6 | 35.8 | 73.1 | 62.2 |
> |           | 6 | 65.7 | 72.8 | 36.4 | 67.4 | 58.3 |
> |           | 8 | 59.3 | 63.9 | 34.0 | 60.8 | 52.1 |
> |           | 10 | 52.3 | 43.2 | 31.6 | 56.2 | 44.1 |
> |           | 12 | 44.2 | 29.5 | 32.2 | 53.1 | 38.4 |
> | MLP Drop | 2 | 75.6 | 75.7 | 39.0 | 74.5 | 65.6 |
> |           | 4 | 66.5 | 70.9 | 36.8 | 69.8 | 59.2 |
> |           | 6 | 72.4 | 71.1 | 40.6 | 71.4 | 62.9 |
> |           | 8 | 61.6 | 66.7 | 36.0 | 66.2 | 55.3 |
> |           | 10 | 53.6 | 59.9 | 34.2 | 61.9 | 49.9 |
> |           | 12 | 50.4 | 49.4 | 30.4 | 60.1 | 45.5 |
> | Attn Drop | 2 | 79.7 | 76.5 | 41.4 | 77.0 | 68.5 |
> |           | 4 | 79.1 | 76.1 | 39.8 | 76.0 | 67.5 |
> |           | 6 | 79.1 | 75.5 | 40.4 | 74.6 | 66.9 |
> |           | 8 | 74.3 | 74.3 | 40.0 | 74.0 | 65.5 |
> |           | 10 | 73.9 | 69.7 | 39.0 | 69.7 | 62.5 |
> |           | 12 | 57.0 | 57.0 | 37.2 | 62.3 | 55.9 |
>
> We will include all these results in the revised version and plan to further expand our evaluation to additional model families in future work.

---

> ### Author Response · Authors · 2025-12-17
> **Response to Reviewer bdwR - Part III**
>
> ## 4. Potential Compute-Intensive Exploration
>
> Thank you for the suggestion. We agree that additional compute-intensive experiments would further strengthen our contribution. In the “Discussions and Conclusion” section, we already acknowledge the current limitations and outline future directions. Due to limited computational resources, we restrict this work to LoRA fine-tuning and leave more compute-heavy training strategies for future exploration.
>
> ---
>
> ## 5. Discussion of Gradient-Related Properties
>
> In this work, we focus on forward-pass properties, specifically how layer dropping affects layer outputs and why certain layers exhibit low importance. Based on the formulation of attention, we attribute low importance to sparse attention patterns, over-smoothed token embeddings, and low-magnitude attention outputs.
>
> Although gradient values reflect parameter update sensitivity, they do not directly indicate inference-time importance. For example, gradient vanishing does not necessarily imply that a layer is unimportant for inference. We plan to further discuss gradient-based metrics in the appendix.
>
> ---
>
> ### References
>
> [1] *ShortGPT: Layers in Large Language Models are More Redundant Than You Expect*, Men et al., arXiv
> [2] *The Unreasonable Ineffectiveness of the Deeper Layers*, Gromov et al., ICLR 2025

---

> > ### Comment · Reviewer_bdwR · 2025-12-26
> > **Response to Authors**
> >
> > Thank you for the thorough replies. Most of my concerns are addressed.

---

> > > ### Author Response · Authors · 2025-12-26
> > >
> > > Thank you for the update, and we appreciate your time and feedback throughout the discussion.

---

### Review · Reviewer_tArF · 2025-11-18

**Summary Of Contributions:**

The paper presents an investigation into the redundancy across different transformer modules. The paper finds that there’s a substantially larger amount of redundancy in the attention modules compared to the MLP. Significant speedup can be achieved by pruning attention layers with only minor performance loss.

**Audience:**

Yes

**Audience Explanation:**

The paper tackles an important question in improving efficiency in transformer models and would be interesting to researchers in this area

**Broader Impact Concerns:**

no concerns

**Claims And Evidence:**

No

**Claims Explanation:**

The pruning baselines are relatively weak -- I provide more details in the Requested Changes.

**Requested Changes:**

Critical:
1. The framing/analyses:
The framing of the paper is a bit unclear. Is this a pruning paper or an interpretability paper? In the current framing, neither aspect is strong enough. For a pruning paper, the baseline of dropping entire transformer blocks is somewhat weak — I’d encourage the authors to discuss SOTA pruning approaches and to compare their approach to the current SOTA. Otherwise, it's very hard to put the performance of the proposed approach into the perspective. If this is an interpretability paper, I’d like to see a deeper analysis of redundancy. Currently, most analyses are validating previous observations.

2. Methods:
a) I’m confused about the Block Drop. P. 2 suggests that this method is proposed by the authors but Table 1 states that it’s adapted from Men et al. ’24. This needs to be clarified in the paper.
b) Table 2: what are reverse order and relative magnitude? These baselines need to be explained so that the reader can understand them without having to look up the provided reference.
c) Section 6.3 — what is the importance score here? Cosine similarity?

3. Results:
a) Table 1: How are the numbers of dropped layers determined? Is there a reason for why the number isn't held constant across all manipulations?

b) Table 4: How was the batch size and the number of layers to drop determined?


Nice-to-have’s:
1. I generally find the results a bit hard to parse since different tables present only a subset of findings. For instance, Table 1 presents the results for Block, Att, and MLP drop for Mistral-7B and Table 2 presents results for the Llama models. However, Table 1 is missing Block and MLP but presents a larger number of Att drop results. The results would be easier to parse if they were consolidated into one table and showed a consistent set of experiments.

---

> ### Author Response · Authors · 2025-12-17
> **Response to Reviewer tArF - Part I**
>
> ## 1. Comparison of Potential Pruning Techniques
>
> Our method operates in a low-complexity search space over the \(L\) layers. This is substantially simpler than traditional weight pruning, whose search space is on the order of \((\text{number of layers} \times \text{number of parameters per layer})\), which is millions of times larger than that of layer dropping.
>
> Therefore, layer dropping does not require sophisticated metrics. We find that simple heuristics already yield near-optimal pruning with minimal performance drop, suggesting that more complex optimization offers limited additional benefit.
>
> To further prove this point, we also compare with more sophisticated metrics such as gradient-based and perplexity-based methods as follows.
>
> We adopt LLaMA-3-8B, as reported in Table 3, and strictly follow the same experimental settings. Among the compared approaches, the cosine-similarity-based method corresponds to our proposed approach, while the gradient-based and PPL-based methods are newly introduced baselines.
>
> For the gradient-based method, we compute gradients on the calibration datasets, average the parameter gradients within each layer, and use the resulting layer-wise gradient magnitude as the dropping metric.
>
> The PPL-based method evaluates the model’s sensitivity to layer removal by measuring performance changes on a calibration dataset. Specifically, we independently drop each attention layer and measure the resulting perplexity. The attention layers are then ranked according to the increase in perplexity from low to high, and this ranking is used to determine the layer dropping priority.
>
> We compare our metric with the aforementioned metrics as follows.
>
> | Drop_n | Metric            | HellaSwag | MMLU | OBQA | Winogrande | Avg. |
> |-------:|-------------------|-----------|------|------|------------|------|
> | 2      | Gradient-based    | 81.7 | 65.3 | 44.6 | 77.5 | 65.4 |
> |        | PPL-based         | 81.7 | 65.2 | 44.6 | 78.1 | 65.5 |
> |        | **Ours (Cosine)** | **82.1** | **65.2** | **45.0** | **78.3** | **67.7** |
> | 4      | Gradient-based    | 81.6 | 65.0 | 44.2 | 78.1 | 65.4 |
> |        | PPL-based         | 82.0 | 65.1 | 45.6 | 78.1 | 65.9 |
> |        | **Ours (Cosine)** | **81.6** | **65.1** | **44.8** | **78.2** | **67.4** |
> | 6      | Gradient-based    | 81.2 | 64.8 | 44.2 | 77.7 | 65.0 |
> |        | PPL-based         | 81.5 | 65.2 | 44.6 | 77.7 | 65.2 |
> |        | **Ours (Cosine)** | **81.8** | **65.2** | **45.4** | **78.8** | **67.8** |
> | 8      | Gradient-based    | 77.7 | 59.5 | 46.0 | 75.1 | 62.7 |
> |        | PPL-based         | 81.0 | 65.2 | 44.4 | 77.7 | 64.9 |
> |        | **Ours (Cosine)** | **81.1** | **65.1** | **45.0** | **78.4** | **67.4** |
> | 10     | Gradient-based    | 76.9 | 59.6 | 44.8 | 74.9 | 62.1 |
> |        | PPL-based         | 78.2 | 59.6 | 43.6 | 74.1 | 61.9 |
> |        | **Ours (Cosine)** | **80.5** | **64.3** | **44.8** | **78.4** | **67.0** |
> | 12     | Gradient-based    | 69.9 | 55.8 | 44.2 | 72.5 | 59.0 |
> |        | PPL-based         | 74.5 | 56.2 | 40.8 | 71.5 | 58.6 |
> |        | **Ours (Cosine)** | **79.4** | **63.9** | **42.2** | **77.8** | **65.8** |
>
> This result shows that cosine similarity performs competitively with more complex metrics. In contrast, our method does not require gradient computation, and cosine similarity can be obtained layer by layer, making it significantly more efficient.

---

> ### Author Response · Authors · 2025-12-17
> **Response to Reviewer tArF - Part II**
>
> ## 2. Baselines of Dropping the Entire Blocks
>
> Existing results already show that dropping entire Transformer blocks severely harms performance, whereas dropping only the attention layers substantially mitigates this degradation. Moreover, the gap between dropping full blocks and dropping attention layers cannot be closed by employing more complex metrics. As noted in our response to W1, the search space for layer dropping is inherently limited, and therefore the potential improvement from more sophisticated metrics is also fundamentally constrained.
>
> As for the comparison against dropping entire Transformer blocks, we have already presented the performance of the widely used ShortGPT in Table 1. We additionally consider *The Unreasonable Ineffectiveness of the Deeper Layers* [2] here.
>
> | Method                                   | ARC-C | BoolQ | HellaSwag | MMLU | OBQA | PIQA | RTE | WinoGrande | Avg. ↑ |
> |------------------------------------------|------|------|-----------|------|------|------|------|------------|--------|
> | ShortGPT (Block-4)                       | 53.1 | 80.4 | 77.5 | 61.6 | 40.0 | 77.6 | 70.0 | 76.6 | 67.1 |
> | Unreasonable Effectiveness (Block-4)     | 53.0 | 80.3 | 76.8 | 61.7 | 40.2 | 77.5 | 69.5 | 76.1 | 66.9 |
> | ShortGPT (Block-8)                       | 40.0 | 71.6 | 63.9 | 60.0 | 30.6 | 69.3 | 63.9 | 69.7 | 58.6 |
> | Unreasonable Effectiveness (Block-8)     | 39.0 | 71.4 | 63.5 | 59.8 | 30.5 | 69.1 | 63.7 | 69.2 | 58.3 |
>
> The performance of ShortGPT and Unreasonable Effectiveness is very close, and our results have already shown that Attention Drop significantly outperforms block dropping. Therefore, we did not include all variants of block-level methods in the main paper. However, we appreciate the reviewer’s suggestion and include additional comparisons here for completeness.
>
> Regarding pruning methods from other perspectives, we have compared a diverse set of baselines, including unstructured pruning methods (such as Wanda) and structured compression techniques (such as block dropping). These baselines cover both training-free and post-training scenarios (e.g., Shortened LLaMA), which represents a finetuned depth-reduced model.
>
> If there are specific additional baselines you recommend, we would be happy to consider them and include further comparisons.
>
> ---
>
> ## 3. Structure of the Paper and Clarification of the Interpretation
>
> Our paper progresses from compression techniques to the discovery of attention redundancy, followed by an empirical interpretation of this phenomenon and the effectiveness of the proposed methods. We believe the structure is clear and accessible, and that our contributions span methodology, empirical findings, and interpretability.
>
> Our interpretation is primarily grounded in training-free empirical analysis. A deeper understanding of attention redundancy would require tracking the training dynamics to pinpoint when and how redundancy emerges and evolves. Due to resource constraints, we are currently unable to conduct such experiments. Nonetheless, we have added a discussion of this topic as a potential avenue for future work.
>
> ---
>
> ## 4. Description of Block Drop
>
> We would like to clarify that the concept of Block Drop is introduced in the Preliminaries section for completeness, and we do not claim it as our proposed method. Our focus lies in the Layer Drop methods, which demonstrate improved performance and robustness under compression compared to prior techniques.
>
> ---
>
> ## 5. Description of Baselines in Table 2
>
> Thanks for pointing this out. We have added the corresponding descriptions in the paragraph for clarification.
>
> ---
>
> ## 6. Importance Scores in Section 6.3
>
> You are right that cosine similarity is the default metric used throughout our paper. We have clarified this after presenting the results in Table 2.
>
> ---
>
> ## 7. Explanation of the Number of Dropped Layers and Batch Size
>
> In Table 1, we progressively drop layers until performance falls below an acceptable threshold (e.g., 95%). Because different types of Transformer blocks and layers exhibit varying degrees of robustness, the number of layers dropped differs across categories. As a result, we omit aggressive dropping ratios for the Block and MLP layers, as their performance degrades substantially at that point. In contrast, attention layers are more redundant and tolerant to removal, so we continue dropping them and report the extended results.
>
> In Table 4, we use the layer configurations that satisfy the acceptable performance threshold. The batch size is progressively increased in powers of two until it reaches the GPU memory limit corresponding to each setting. For the 70B model, we use 8 NVIDIA A6000 Ada GPUs, while for models of other sizes, we use 4 NVIDIA A6000 Ada GPUs.
>
> ---
>
> ## References
>
> [1] *ShortGPT: Layers in Large Language Models are More Redundant Than You Expect*, Men et al., arXiv
> [2] *The Unreasonable Ineffectiveness of the Deeper Layers*, Gromov et al., ICLR 2025

---

> > ### Author Response · Authors · 2025-12-31
> > **Follow-up on our responses to your comments**
> >
> > Dear Reviewer tArF,
> >
> > We hope you are doing well. We would like to kindly follow up regarding our previous response to your comments. We have provided additional analyses, new experiments, and detailed clarifications, particularly on the comparison of pruning metrics, block-level baselines, paper structure, interpretation discussion, and the explanation of experimental settings.
> >
> > We would really appreciate it if you could let us know whether our current response sufficiently addresses your concerns, or if any further clarification or additional evidence would be helpful. We are happy to provide more experiments or discussion if needed.
> >
> > Thank you again for your valuable feedback and your time.
> >
> > Best regards,
> > Submission6033 Authors

---

> ### Author Response · Authors · 2026-01-12
>
> Dear Reviewer tArF,
>
> We apologize for the additional follow-up. As the discussion period is approaching its end, we would like to gently check in to see whether our previous responses have adequately addressed your concerns.
>
> If there are any remaining questions or points that would benefit from further clarification, we would be very grateful to address them.
>
> Thank you again for your time and consideration.
>
> Best regards,
> Submission6033 Authors

---

> > ### Comment · Reviewer_tArF · 2026-01-12
> >
> > Thank you for providing these clarifications and additional baselines.
> >
> > My take-home message after reading the responses is that your method is substantially simpler compared to the existing approaches while providing comparable performance. After reading your responses, I agree with you.
> >
> > The issue is that as a non-expert in pruning, I didn't get it from the paper alone. For instance, I missed the comparisons to Shortened LLaMA and Wanda since they were hidden in the appendix. And even then, these approaches weren't explained in enough detail for me to understand how they compare to the proposed approach -- I had to look them up in the respective papers, which I shouldn't need to do as the paper should stand on its own.
> >
> > That said, I'm persuaded by the findings but I do think small revisions in the presentation would be beneficial for the diverse readership of TMLR: moving the baselines to the main text and explaining what they are and how they differ from the proposed approach would help clarify the take-home message.

---

> > > ### Author Response · Authors · 2026-01-13
> > >
> > > Thank you for the thoughtful feedback and for carefully reading our responses. We appreciate your perspective and are glad that the take-home message of our work is now clearer.
> > >
> > > Following your suggestion, we have revised the paper by moving the comparisons with Shortened LLaMA and Wanda from the appendix to the main text. We also added concise descriptions of these baselines, explaining their key ideas and how they differ from our proposed approach, so that readers can grasp the essential concepts of these methods without consulting the original papers.
> > >
> > > We believe these revisions substantially improve the clarity and accessibility of the paper for a broad TMLR audience. Thank you again for the constructive suggestions, which have helped us improve the presentation of our work.

---

### Review · Reviewer_pEvQ · 2025-12-09

**Summary Of Contributions:**

Authors explore redundancy in transformer models via block, MLP, and attention modules and find attention has a surprising redundancy. With this property, the authors designed Attention Drop which improves efficiency without sacrificing accuracy. The paper shows that transformer models have certain redundancy and have a lot of spares for improving performance.

**Audience:**

Yes

**Audience Explanation:**

A wide range of audiences would be interested in the findings of the paper as redundancy can help instruct the design of new efficient LLM and transformer architectures. The findings over KV-cache efficiency improvement are also important.

**Broader Impact Concerns:**

No ethical concerns need to get addressed.

**Claims And Evidence:**

Yes

**Claims Explanation:**

Most contributions in the paper were supported by analysis and experimental results. However, I have some further comments:

1. A fundamental question the authors neglect is whether KVCache can keep the model sustainable. Authors may neglect one impact of scaling law and sparsity. As we can see from LoRA and other examples, sparse neural networks are easier for post-training and instruction tuning. Further training can embed more knowledge easier. To show attention drop is effective, one important study is that of investigating that after attention drop and than conducting post-training tasks and seeing how performance is.

2. Authors show that attention has sparsity and redundancy but this does not directly correlate to inefficiency. For example, DeepSeek-V2 / V2.5 leverages sparse attention to improve the time complexity of computation of attention from O(n^2) to O(nk) where k<<n. In other words, having attention sparsity may improve efficiency.

**Requested Changes:**

Methods:
1. I am a little bit confused by the block-drop. How do you calculate the cosine similarity. Is it the average for all samples where you inference all X with the model? If so, how many samples do you use? If not, how do you make sure that the cosine similarity results apply to other data samples?
2. Any theoretical analysis of why cosine similarity is a good metric? For example, let's say for an MLP layer X'=MLP(X). If cos(X', X) is small, what is the relationship with the redundancy of this layer of MLP?

Experiments:
1. Experiments show that attention drop will not affect post-training.
2. Authors should show either attention drop is orthogonal to sparse attention and can work with sparse attention (e.g. applied to an open sourced DeepSeek sparse attention model) or prove that for the same backbone model, attention drop is more efficient than sparse attention. I think the former one is easier as usually we don't have enough compute resources to do the later experiments.

Writing:
1. (Minor). The fonts in figure 1,7,8,9,10,11 are too small.

Optional:
1. The paper studied block, MLP, and attention. I am quite interested in whether MOE will contain any redundancy as experts are supposed to be designed for efficient usage of knowledge in neural networks.

---

> ### Author Response · Authors · 2025-12-17
> **Response to Reviewer pEvQ - Part I**
>
> We sincerely thank the reviewer for the insightful comments and questions. In the following, we address each point in turn and provide clarifications.
>
> ## 1. The Impact of KV-Cache on Scaling and Sparsity
>
> Thank you for raising this point. Our primary goal is to investigate redundancy within Transformer architectures, and we do not overlook the role of the KV-Cache. Since our method removes entire attention layers without modifying the attention mechanism itself, the KV-Cache for all retained layers remains unchanged. The reduction in KV-Cache memory naturally follows from dropping the corresponding attention layers, rather than altering how KV states are computed or stored.
>
> ---
>
> ## 2. Post-Finetuning Performance after Attention Drop
>
> Regarding post-training, we have already included results in Tables 15 and 16. These results show that models with Attention Drop, after lightweight LoRA fine-tuning, achieve performance comparable to their non-compressed counterparts. This supports the redundancy of attention layers and demonstrates that the compressed models remain trainable and recoverable, further validating the effectiveness of our approach.
>
> ---
>
> ## 3. Correlation Between Sparsity, Redundancy, and Inefficiency
>
> Thank you for the thoughtful comment. We would like to clarify the relationship between sparsity, redundancy, and inefficiency.
>
> In our context:
> - **Sparsity** refers to the uneven contribution of different weights or layers, where some components play an important role while others contribute very little.
> - **Redundancy** focuses on components whose outputs have minimal impact on the model’s behavior.
>
> In both cases, certain computations provide negligible contribution to the final output, and performing these computations nonetheless results in inefficiency.
>
> Our findings reveal redundancy at the **layer level**, showing that some attention layers can be entirely removed. This does not conflict with **intra-layer sparse attention**, which reduces computation within an attention layer. Instead, it highlights that redundancy exists at multiple granularities.
>
> Moreover, our analysis suggests that intra-layer sparsity often leads to small output changes, which in turn contributes to layer-level redundancy. When an attention layer already performs highly sparse or low-magnitude updates, its overall transformation becomes negligible, making the entire layer removable. This connection further supports our findings and explains why layer dropping is effective.
>
> ---
>
> ## 4. Clarification on Cosine Similarity Calculation for Block Drop
>
> Thank you for the question. We compute cosine similarity by measuring the similarity between each layer’s input and output representations over a small calibration dataset. Specifically, for each sample, we obtain the input `x` and output `y`, compute `cos(x, y)`, and average these scores across samples to obtain the layer’s importance.
>
> We use 256 calibration samples (Appendix A), consistent with training-free pruning practices. To ensure generalization, Figure 16 in Appendix C reports extensive ablations across different datasets (C4, WikiText, Alpaca) and different calibration sizes (4–1024 samples). The resulting importance rankings remain highly consistent, and the selected layers to drop hardly change. This shows that cosine similarity captures intrinsic layer behavior rather than dataset-specific patterns, making a small calibration set both sufficient and reliable.

---

> > ### Author Response · Authors · 2025-12-17
> > **Response to Reviewer pEvQ - Part II**
> >
> > ## 5. Clarification on the Use of Cosine Similarity
> >
> > Thank you for the question. We would first like to correct an important misunderstanding in the formulation. The reviewer’s example assumes an MLP layer computes X' = MLP(X). However, in Transformer architectures, every sublayer (including MLP) is wrapped inside a residual connection, so the actual output is Y = X + MLP(X).
> >
> > This formulation is explicitly stated in our paper and further illustrated in Appendix C (Residual Connection). Therefore, cosine similarity is computed between `X` and `Y`, not between `X` and `MLP(X)`.
> >
> > This metric has also been used in prior block-dropping works such as **ShortGPT** [1]. Under the residual form `y = x + f(x)`, the cosine similarity between `x` and `y` reflects both the direction and the magnitude of `f(x)`. A large residual update reduces similarity, while extremely high similarity implies that `f(x)` is both small and well aligned with `x`.
> >
> > Empirically, as shown in Figures 3 and 10, many attention layers exhibit cosine similarity above 0.95 or even 0.99, indicating that their residual updates are negligible and the layers are functionally redundant. Figure 13 further shows that attention outputs have consistently lower magnitude than MLP outputs, especially in deeper layers, which aligns with the cosine similarity trend.
> >
> > While cosine similarity may not be the most sophisticated metric, it is highly effective for layer-level pruning, where the search space is small (only L layers) and the improvement space is limited. In such a setting, simple and training-free heuristics can still yield strong performance, as demonstrated by our pruned models.
> >
> > ---
> >
> > ## 6. Post-Training After Attention Drop
> >
> > Yes, we have already conducted the post-training experiments requested by the reviewer. As shown in Tables 15 and 16, models after Attention Drop can recover most of their performance through lightweight LoRA fine-tuning, achieving accuracy comparable to or even matching the non-compressed baseline. These results demonstrate that attention dropping does not hinder post-training and that compressed models retain strong learnability.
> >
> > ---
> >
> > ## 7. Relationship Between Attention Drop and Sparse Attention
> >
> > Thank you for the insightful comment. Attention Drop and sparse attention address efficiency at different levels and therefore do not conflict with each other.
> >
> > ### 8. Different Granularities of Efficiency
> > - **Sparse attention** improves efficiency within a layer by reducing the number of attended tokens.
> > - **Attention Drop** improves efficiency at the layer level by removing entire attention layers.
> >
> > These mechanisms target distinct granularities and are conceptually orthogonal.
> >
> > ### 9. Layer-Level Redundancy Complements Intra-Layer Sparsity
> > Our analysis shows that some attention layers produce extremely small or negligible updates. When a whole layer is redundant, removing it is strictly more efficient than sparsifying its operations, as it eliminates all FLOPs and KV-cache updates for that layer. Attention Drop can thus be viewed as a more aggressive efficiency strategy applied to less important attention layers.
> >
> > ### 10. Compatibility
> > Since our method does not modify the internal structure of retained layers, it is fully compatible with sparse-attention mechanisms. A model can first prune redundant layers and then apply sparse attention within the remaining layers without interference.
> >
> > ---
> >
> > ## 11. Font Size in Figures
> >
> > Thank you for pointing this out. We agree that the font sizes in Figures 1, 7, 8, 9, 10, and 11 were relatively small. We have updated all figures with larger and more readable fonts in the revised version to improve clarity.
> >
> > ---
> >
> > ## 12. Expert Redundancy in MoE Architectures
> >
> > Redundancy in MoE architectures is an interesting and important topic. Expert pruning in MoE models typically operates at a finer granularity, as experts correspond to sublayers within MLP blocks, whereas our work focuses on redundancy at the layer level in dense Transformers. Therefore, MoE redundancy falls outside the scope of this paper.
> >
> > Nevertheless, existing research has shown that MoE models do contain notable redundancy. Several studies demonstrate that some experts can be removed or skipped for certain tasks without significantly affecting performance [2,3]. These findings suggest that redundancy exists at multiple levels within Transformer architectures and complement our observations.
> >
> > ---
> >
> > ## References
> >
> > 1. ShortGPT: Layers in Large Language Models are More Redundant Than You Expect, Men et al., arXiv 2024
> > 2. Towards Efficient Mixture of Experts: A Holistic Study of Compression Techniques, He et al., TMLR
> > 3. Not All Experts Are Equal: Efficient Expert Pruning and Skipping for Mixture-of-Experts Large Language Models, Lu et al., ACL 2025

---

> > > ### Comment · Reviewer_pEvQ · 2025-12-18
> > >
> > > Thank the authors for the clarification especially on how cos similarity is calculated. I think it is now much clearer.
> > >
> > > A follow-up question:
> > >
> > > For post-training tasks, I think I would refer to doing some post-training tasks directly on the model with layer-drop and attention drop-algorithms.
> > >
> > > Let's say you have a pre-trained Llama model (no instruction tuning, only pertaining): meta-llama/Meta-Llama-3-8B (Base pre-trained model). We call it M1. Then we apply layer-drop to remove redundancy and we call the resulting model M2. In real-world, we will want to distribute M2 because it is more light-weighted and efficient:
> > >
> > > Next you do three groups of comparison:
> > > 1. Do continued pre-training on a domain knowledge dataset for M1 and M2 and compare perplexity or next word prediction.
> > > 2. Instruction tuning on Alpaca for M1 and M2 and see accuracy of task competence/instruction following
> > > 3. Instruction tuning+RLHF/DPO on specific tasks dataset for M1 and M2 (e.g. math, coding, HotPotQA) and see accuracy.
> > > You can either do post-finetuing with LoRA or no fine-tuning after layer drop. But I think no fine-tuning and directly do instruction tuning will make the results stronger.
> > >
> > > Do you have similar experiments? Please correct me if I may have missed something in the paper.

---

> > > > ### Author Response · Authors · 2025-12-18
> > > > **Clarification of the Study Scope**
> > > >
> > > > Thank you for the follow-up question. We would like to clarify that our paper focuses on a **training-free analysis of structural redundancy**. As supportive evidence, we include only lightweight LoRA post-training in Appendix **Tables 15 and 16**, which serves as a supplementary verification of the discussions in *Discussions and Conclusion*. These results demonstrate that the compressed models remain trainable and can recover performance with minimal computational cost.
> > > >
> > > > The experiments you proposed, including full continual pretraining, full instruction tuning, and RLHF or DPO after layer dropping, fall under a **different problem setting**. These procedures require retraining the entire model and involve substantial computational resources. Their purpose is to investigate retraining dynamics after compression, rather than redundancy or training-free model compression. For this reason, such experiments are outside the scope and feasibility of the present study.

---

> > > > > ### Comment · Reviewer_pEvQ · 2025-12-23
> > > > >
> > > > > I see. Then I think your scope and focus is over LLM which are instruct tuned rather than general LLM (pre-trained or continue pre-trained). For example, you are focusing on Llama3-70B-Instruct rather than Llama3-70B. I think this makes sense to me and still makes the paper have a good argument. But can you revise your paper to clarify the scope, especially the abstract, first section of methodology and experiment? The model name listed in the paper also needs to be clarified, e.g. Llama3-70B-Instruct.

---

> > > > > > ### Author Response · Authors · 2025-12-25
> > > > > >
> > > > > > Thank you for the helpful suggestion. We would like to clarify that our study is not limited to base language models. Our experiments include both base models and instruction-tuned variants. For example, we evaluate Llama-3-8B (base), Llama-3-8B-Instruct, Mistral-7B-Math, and Llama-2-7B-Math, and we observe similar redundancy patterns across these different model types.
> > > > > > Thank you again for pointing this out, we will also standardize the naming of models throughout the paper to avoid ambiguity to improve clarity.

---

### Author Response · Authors · 2025-12-21
**Follow-up on the Author–Reviewer Discussion**

Dear Reviewers,

We hope you are doing well. We would like to kindly check whether our recent responses have addressed your earlier concerns. If any points remain unclear or would benefit from further clarification, please let us know, and we will be happy to provide additional details.

As the discussion period is drawing to a close, we would also appreciate any further feedback that could help us improve the paper or support your final assessment.

Thank you again for your time and for engaging with our submission.

Best,
Authors of Submission 6033

---

> ### Author Response · Authors · 2025-12-23
>
> Dear Reviewers,
>
> We would like to kindly follow up and check whether there are any remaining questions, concerns, or points where additional clarification would be helpful. If there are aspects that would benefit from further explanation or additional evidence, we would be very happy to provide them promptly.
>
> As the discussion period is approaching its end, any further feedback or guidance would be greatly appreciated, as it would help us strengthen the paper and support your final evaluation.
>
> Thank you again for your time and thoughtful engagement with our submission.
>
> Best regards,
> Authors of Submission 6033

---

### Decision · Action_Editor_UfA1 · 2026-01-24

**Recommendation:** Accept with minor revision

**Additional Comments:**

One of the reviewers pointed that the authors need to provide more evidence that dropping operations will not affect post-training performance. I think authors should spend some effort either stating a very clear scope in the paper or complementing experiments for base models.

**Audience:**

Yes

**Audience Explanation:**

Yes, paper is of interest to TMLR community.

**Claims And Evidence:**

Yes

**Claims Explanation:**

Two reviewers think that claims in this paper are supported through analysis and experiments.

One reviewer had concerns for which the authors need to make some revisions.

---

> ### Author Response · Authors · 2026-02-06
> **Response to AE Decision Summary**
>
> Dear Action Editor,
>
> Thank you very much for the decision and for summarizing the reviewers’ feedback.
>
> Regarding the remaining concern about whether dropping operations affect post-training performance, we would like to note that we have now included additional post-training results for both base pretrained models and instruction-tuned models. These results are presented in Figure 15 and in the section “Post-Training on Compressed Models” in Appendix B. They demonstrate that attention-dropped models remain fully trainable and can effectively recover performance through subsequent post-training, thereby supporting the feasibility of our approach.
>
> We have also expanded the Discussions and Conclusion section to explicitly incorporate these findings, articulate their implications, and clarify how they motivate and guide potential future research directions.
>
> Thank you again for your time and guidance.
>
> Best regards,
> TMLR Paper6033 Authors